# DNA replication in primary hepatocytes without the six-subunit ORC

**Róża K Przanowska[1†], Yuechuan Chen[2†], Takayuki-Okano Uchida[3], Etsuko Shibata[2], Xiaoxiao Hao[2], Isaac Segura Rueda[2], Kate Jensen[1], Piotr Przanowski[1], Anthony Trimboli[3], Yoshiyuki Shibata[2], Gustavo Leone[3], Anindya Dutta[1,2]\***

[1]Dept. of Biochemistry and Molecular Genetics, University of Virginia, Charlottesville, United States; [2]Dept. of Genetics, University of Alabama at Birmingham, Birmingham, United States; [3]Cancer Center, University of Wisconsin in Milwaukee, Milwaukee, United States

**\*For correspondence:**
Duttaa@uab.edu

[†]These authors contributed equally to this work

## eLife Assessment

This **valuable** descriptive manuscript builds on prior research showing that the elimination of Origin Recognition Complex (ORC) subunits does not halt DNA replication. The authors obtain **solid** data using various methods to genetically remove one or two ORC subunits from specific tissues and still observe replication. The replication appears to be primarily endoreduplication, indicating that ORC-independent replication may promote genome reduplication without mitosis. The mechanism behind this ORC-independent replication remains to be elucidated. The study and mutants described herein lay the groundwork for future research to explore how cells compensate for the absence of ORC and to develop functional approaches to investigate this process. The reviewers suggested the observations could be supported by additional experiments. This work will be of interest to those studying genome duplication and replication.

**Abstract** The six-subunit ORC is essential for the initiation of DNA replication in eukaryotes. Cancer cell lines in culture can survive and replicate DNA replication after genetic inactivation of individual ORC subunits, ORC1, ORC2, or ORC5. In primary cells, ORC1 was dispensable in the mouse liver for endo-reduplication, but this could be explained by the ORC1 homolog, CDC6, substituting for ORC1 to restore functional ORC. Here, we have created mice with a conditional deletion of ORC2, which does not have a homolog. Although mouse embryo fibroblasts require ORC2 for proliferation, mouse hepatocytes synthesize DNA in cell culture and endo-reduplicate in vivo without ORC2. Mouse livers endo-reduplicate after simultaneous deletion of ORC1 and ORC2 both during normal development and after partial hepatectomy. Since endo-reduplication initiates DNA synthesis like normal S phase replication these results unequivocally indicate that primary cells, like cancer cell lines, can load MCM2-7 and initiate replication without ORC.

## Introduction

DNA replication during normal mitotic cycles or during endo-reduplication is initiated by the six-subunit ORC, five of which have AAA+ATPase domains and form a ring-shaped complex that binds DNA and bends it (**Bell and Stillman, 1992**; **Neuwald et al., 1999**; **Dhar et al., 2001a**; **Clarey et al., 2006**; **Bleichert et al., 2015**). In cooperation with another ATPase, CDC6, and with CDT1, ORC helps load MCM2-7 double hexamers at or near the origins of replication (**Costa and Diffley, 2022**; **Stillman, 2022**; **Hu and Stillman, 2023**). To initiate replication (or endo-reduplication), the MCM2-7 hexamer associates with additional proteins to form the functional CMG helicase that is essential for

unwinding the double-stranded chromosomal DNA, so that the resulting single-stranded DNAs can serve as templates for copying by DNA polymerases. In the lower eukaryotes, various ORC subunits have been consistently found to be essential for DNA replication and cell proliferation (*Bell et al., 1993*; *Foss et al., 1993*; *Micklem et al., 1993*; *Loo et al., 1995*; *Semple et al., 2006*). In mammalian cells, knockdown of individual ORC subunits has also been reported to stop cell proliferation (*Chou et al., 2021*; *Prasanth et al., 2004*). In humans, mutations in *ORC1*, *ORC4*, *ORC6*, *CDT1*, and *CDC6* cause microcephalic primordial dwarfism resembling Meier-Gorlin syndrome (*Bicknell et al., 2011b*; *Bicknell et al., 2011a*; *Guernsey et al., 2011*).

There have, however, been isolated reports that suggest that under unusual circumstances, DNA replication can initiate in eukaryotes in the absence of the full ORC. First hypomorphic mutation made in the *ORC2* gene in HCT116 colon cancer cells decreased ORC2 protein levels by 90% and cells were still viable and could proliferate (*Dhar et al., 2001b*). Deletion of the *Orc1* gene in *Drosophila* permitted several rounds of replication in the resulting larvae and pupae (*Park and Asano, 2008*). A few human cancer cell lines have been created by CRISPR-Cas9 mediated genome engineering where ORC1, ORC2, and ORC5 cannot be detected by regular immunoblots and yet the cell lines survive, proliferate and replicate DNA with the normal complement of origins of replication (*Shibata et al., 2016*; *Shibata and Dutta, 2020*). However, the *Orc2* gene is essential for viability and in the *Orc2Δ* cell line a very minimal level (<0.1% of wild-type levels) of a truncated protein can be detected that reacts with anti-ORC2 antibody and co-immunoprecipitates with ORC3 (*Chou et al., 2021*). Given that WT cells have about 150,000 molecules of ORC2, even if this truncated protein is functional ORC2, ~150 molecules of the protein would be expected to load MCM2-7 double hexamers on at least 50,000 origins of replication. This possibility would be consistent with current models of MCM2-7 loading only if ORC was highly catalytic and one ORC hexamer was capable of loading ~667 MCM2-7 hexamers (*Costa and Diffley, 2022*; *Stillman, 2022*; *Hu and Stillman, 2023*).

To address whether the entire ORC holocomplex is essential for DNA replication initiation in primary mammalian cells, we have reported a conditional mutation in the ORC1 gene of mouse (*Okano-Uchida et al., 2018*). Conditional deletion of *Orc1* (using tissue-specific Cre drivers) revealed that the gene was essential for normal mitotic cell division in intestinal epithelial cells, but that endoreduplication in tissues that are known to become polyploid, like placental trophoblasts and the liver hepatocytes, was unaffected. Because the ORC1 protein is a functional ATPase with sequence homology with CDC6 (*Saha et al., 1998*; *Williams et al., 1997*), one possibility that remained was that the mouse CDC6 protein substituted for ORC1 in the ORC holocomplex, producing enough functional ORC. Therefore, we decided to make a conditional mutation in another subunit of ORC, *Orc2*, that does not have the Walker A or B motifs that would make it a functional ATPase and that is not homologous to *Cdc6*. We designed the mutation such that even if an N terminally truncated protein is expressed from a downstream methionine from an alternately spliced mRNA the AAA + like domain will suffer a significant deletion.

This will allow us to test whether the mouse liver survival and endoreduplication after loss of ORC1 is seen even after loss of ORC2. It will also test whether combining the conditional mutations of ORC1 and ORC2 will allow the liver cells to survive and endoreduplicate. If hepatocyte survival and endoreduplication is seen in these mice, then it would be hard to explain this by the substitution of two ORC subunits by the CDC6 protein, and we would gain support for the hypothesis that in rare circumstances even untransformed mammalian cells are capable of loading enough MCM2-7 in the absence of ORC to permit extensive DNA replication.

## Results

### ORC2 is essential for embryonic development and proliferation of mouse embryo fibroblasts

Mice with LoxP sites inserted flanking exons 6 and 7 of mouse *Orc2* were purchased from Cyagen Biosciences Inc (*Figure 1A*). Genotyping with primers F2 and R2 distinguished the loxP-marked allele from the WT allele (*Figure 1B*). In a cross of *Orc2^{f/+}* mice, there was no significant decrease in the yield of *Orc2^{f/+}* or *Orc2^{f/f}* compared to *Orc2^{+/+}* mice (*Figure 1C*), suggesting that the insertion of the loxP sites in the introns of *Orc2* did not impair the function of ORC2. Recombination between the loxP sites would delete the coding exons 6 and 7 of *Orc2*, which removes amino acids L111-V150 (*Figure 1D*).

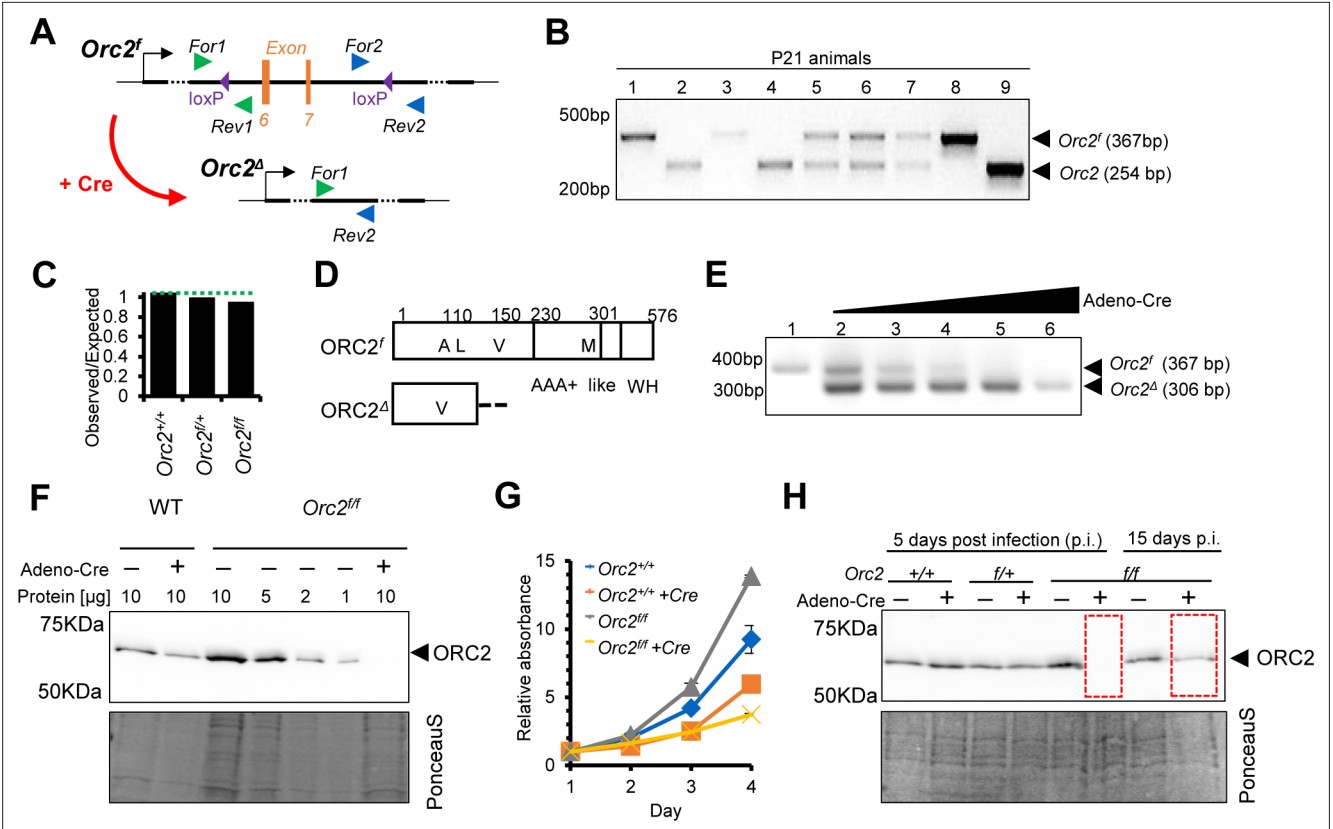

**Figure 1.** Generation of *Orc2^f/f* mice and ORC2 knockout (KO) mouse embryo fibroblasts (MEFs). (**A**) Scheme of introduced loxP sites in *Orc2* locus. (**B**) Representative picture of genotyping of offspring coming from *Orc2^f/+* crossed with *Orc2^f/+*. (**C**) The ratio of observed to expected animals coming from *Orc2^f/+* crossed with *Orc2^f/+*. (**D**) Schematic of the ORC2 protein and the DeltaORC2 protein produced after deletion of exons 6 and 7. A110 is mutated to V110 and then the protein goes out of frame. (**E**) Validation of *Orc2* deletion 3 d after Adeno cre transduction. (**F**) Western blot of ORC2 protein 5 d after Adeno cre transduction. 10 or indicated µl of lysate loaded/lane as written on the top. (**G**) MTT assay of WT and *Orc2^f/f* MEFs without and with Adeno cre transduction. (**H**) Western blot of ORC2 protein 5 and 15 d after Adeno Cre transduction.

The online version of this article includes the following source data for figure 1:

**Source data 1.** PDF file containing original DNA gel picture corresponding to *Figure 1*, panel B, indicating the relevant bands and individual animals.

**Source data 2.** Original image for *Figure 1*, panel B.

**Source data 3.** PDF file containing original DNA gel picture corresponding to *Figure 1*, panel E, indicating the relevant bands and increasing Adeno-Cre.

**Source data 4.** Original image for *Figure 1*, panel E.

**Source data 5.** PDF file containing original Western blot membrane picture corresponding to *Figure 1*, panel F, indicating the relevant bands and addition of Adeno-Cre.

**Source data 6.** Original image for *Figure 1*, panel F.

**Source data 7.** PDF file containing original Western blot membrane picture corresponding to *Figure 1*, panel H, indicating the relevant bands and ORC2 protein expression.

**Source data 8.** Original image for *Figure 1*, panel H.

If a transcript is expressed that skips exons 6 and 7, then the resulting protein mutates A110 to V110 and then throws the protein-coding sequence out of frame (*Figure 1D*, ΔORC2), so that the bulk of the 576 amino acid ORC2 protein, including the AAA + like and WH domains (K230-A576) are not expressed. The AAA + like domain and the WH domain are key elements for ORC2 assembly into ORC and for ORC function. Even in the remote possibility that a truncated protein is expressed due to alternative splicing and translation initiation from an internal methionine, the next methionine is at M301, so that half of the protein including 70 amino acids of the 239 amino acid AAA+-like domain (K230-G469) would be deleted.

**Table 1.** Embryonic lethality of *Orc2 KO*.
The *Orc2Δ* allele was created by expressing Cre recombinase from a *Sox2* promoter in the *Orc2^f/f* embryos.

Offspring from *Orc2^{+/Δ}* intercrosses

| Stage | Genotype | | | Empty decidua | total # (litters) |
|---|---|---|---|---|---|
| | *Orc2+/+* | *Orc2+/Δ* | *OrcΔ /Δ* | | |
| E3.5 | 18 | 30 | 6 | 6 (n.d) | 60 (6) |
| E7.5 | 9 | 21 | 0 | 16 | 46 (5) |
| E13.5 | 9 | 31 | 0 | 15 | 55 (6) |
| 2 wk | 52 | 115 | 0 | - | 167 (27) |

The *Sox2-Cre* allele expresses active Cre during embryonic development. Crossing the *Orc2^{f/f}* mice with *Sox2-Cre* mice resulted in no *Orc2^{Δ/Δ}* embryos at E7.5 days onwards suggesting that embryonic deletion of *Orc2* is lethal (*Table 1*). The near expected number of *Orc2^{+/Δ}* embryos suggests that hemizygosity of *Orc2* can still support viability.

Mouse embryo fibroblasts (MEF) were obtained from *Orc2^{f/f}* E13.5 day embryos and cultured in vitro. Three days after infection with an Adenovirus expressing Cre, effective recombination between the loxP sites is seen, resulting in a genotype showing that exons 6 and 7 have been deleted in most of the MEF (*Orc2^Δ*) (*Figure 1E*). Consistent with this, at 5 d after adeno-Cre infection, the ORC2 protein is not detected in the MEF population (*Figure 1F*). The dilution of the undeleted MEF lysate on the same blot suggests, that if any ORC2 protein is expressed in the MEF after adeno-Cre infection, it is <10% of that in the undeleted MEF. The proliferation rate of the MEF were measured by MTT assays. Even in the *Orc2^{+/+}* MEF, the infection with adeno-Cre decreased proliferation a little (the orange line compared to the blue line in *Figure 1G*). However, for *Orc2^{f/f}* MEF infection with adeno-Cre impairs proliferation even further (yellow line compared to black line in *Figure 1G*). Furthermore, when the MEF were cultured for 15 d, the surviving cells express significant levels of ORC2 (*Figure 1H*), suggesting that the MEFs that had not undergone the Cre-mediated deletion take over the culture.

Taken together, these results suggest that Cre-mediated deletion of exons 6 and 7 of *Orc2* leads to early embryonic lethality and impairs the proliferation of normal diploid MEFs in culture.

## Knockout of *Orc2* in developing mouse liver makes ORC2 protein undetectable in hepatocytes and yet supports most of normal development and endoreduplication

Mice carrying one copy of the Albumin promoter-driven Cre gene (*Alb-Cre*) express the Cre recombinase specifically in hepatocytes (*Postic et al., 1999*). When the *Orc2^{f/f}* mice are crossed with mice carrying *Alb-Cre* (shortened below as *Alb*), the Cre recombinase is expected to promote the recombination-mediated deletion of exons 6 and 7 of *Orc2* in the hepatocytes. The *Alb^{+/-}-Orc2^{f/f}* mice were viable and fertile (*Figure 2A and B*). Crossing such mice yielded the expected numbers of *Alb^{+/-}-Orc2^{f/f}* mice, though there was a drop in the yield of *Alb^{+/+}-Orc2^{f/f}* mice. One allele of *Alb-Cre* is sufficient to express enough Cre to carry out homozygous deletion of the *Orc2* allele in the hepatocytes, so that the results suggest that deletion of *Orc2* in the hepatocytes did not impair viability. The partial lethality of the *Alb^{+/+}* (HOM) mice could be due to the toxicity of high dose of Cre recombinase expressed, something that has been noted by other groups (*Schmidt et al., 2000*; *Loonstra et al., 2001*; *Schmidt-Supprian and Rajewsky, 2007*; *Naiche and Papaioannou, 2007*; *Janbandhu and Moik, 2014*). Because the Alb-Cre is expressed only in the hepatocytes and not in other cells in the liver, we isolated hepatocytes from these livers by growing them in culture to determine whether the ORC2 protein was decreased (*Figure 2C*). The ORC2 protein was significantly decreased in all five of the *Alb^{+/-}-Orc2^{f/f}* mice, showing consistent and near 100% effect of the deletion in the hepatocytes. Western blots for several other ORC subunits, and CDC6 protein showed that the loss of ORC in the same hepatocytes did not significantly decrease these proteins, though MCM2 and MCM3 (two subunits of the MCM2-7 helicase component) were decreased by 50% (*Figure 2D*). The body weights of the *Alb, Orc2^{f/f}* mice were smaller than in the *Orc2^{f/f}* animals of both sexes, though the liver size and liver size normalized to body weight was significantly smaller only in the females (*Figure 2E–G*).

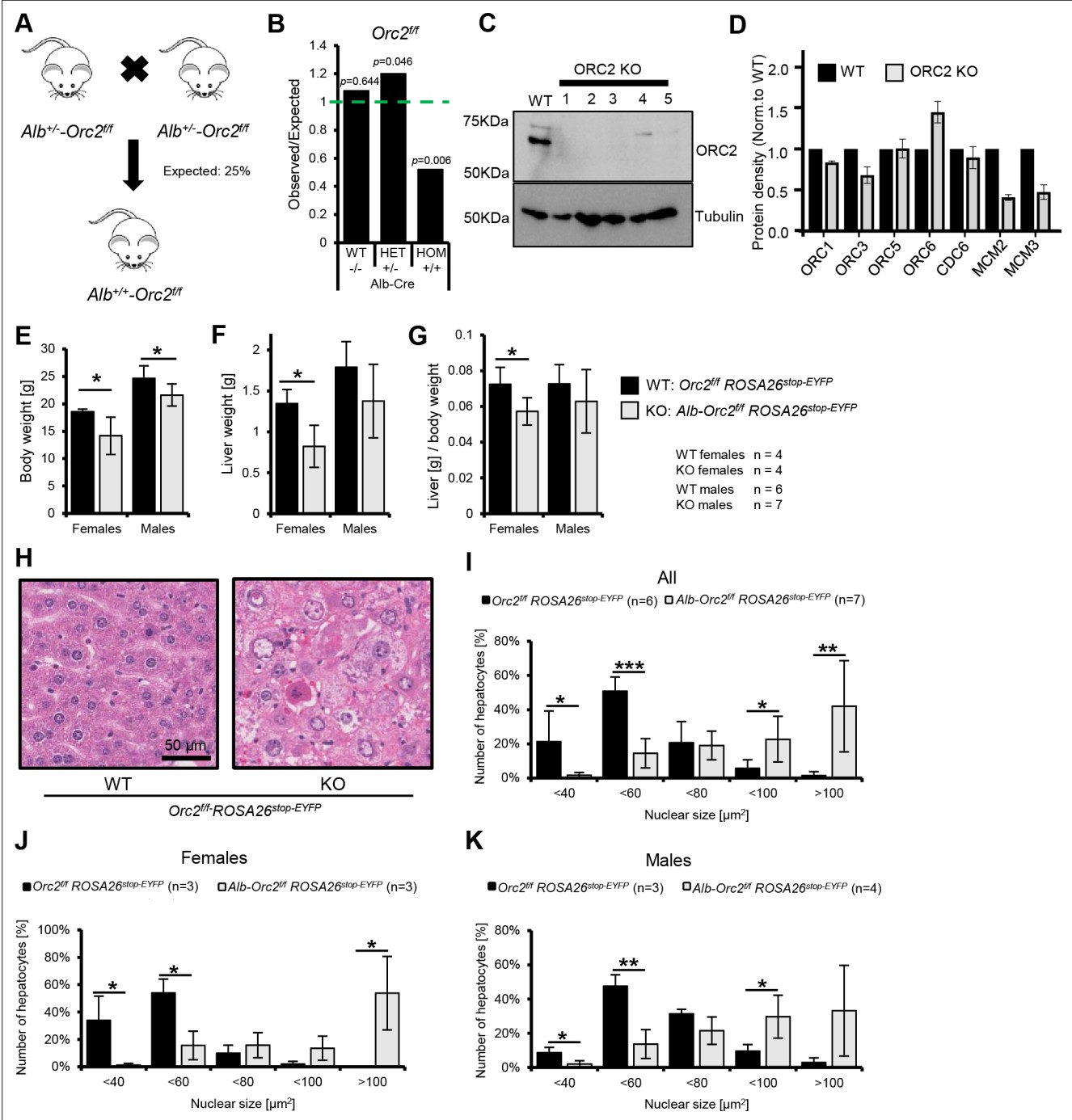

**Figure 2.** Development of liver in *Orc2* KO mice. (**A**) Scheme of *Alb⁺/⁻-Orc2^f/f ROSA26^stop-EYFP* crossed with *Alb⁺/⁻-Orc2^f/f ROSA26^stop-EYFP* (All mice are with *ROSA26^stop-EYFP* and so we do not include this in the genotypes below). (**B**) The ratio of observed to expected animals coming from A. (**C**) Western blot of hepatocytes from *Orc2^f/f* and Alb⁺/⁻-*Orc2^f/f* animals. Tubulin was used as loading control. (**D**) Quantification of the Western blots of hepatocyte lysates from *Orc2^f/f* (without *Alb-cre*) mice and the same genotype but with *Alb-Cre* to show the levels of other key replication initiation proteins in the ORC2 KO hepatocytes. (**E**) Average body weight of *Orc2^f/f* and *Alb-Orc2^f/f* animals. (**F**) Average liver weight of *Orc2^f/f* and *Alb-Orc2^f/f* animals. (**G**) Average liver-to-body weight ratio of *Orc2^f/f* and *Alb-Orc2^f/f* animals. (**H**) Representative H&E staining of liver tissue from *Orc2^f/f* (WT) and *Alb-Orc2^f/f* (KO) animals. Both panels at same scale. (**I**) Quantification of hepatocyte nuclear size in *Orc2^f/f* and *Alb-Orc2^f/f* animals. (**J**) Quantification of hepatocyte nuclear size in *Orc2^f/f* and *Alb-Orc2^f/f* female mice. (**K**) Quantification of hepatocytes nuclear size in *Orc2^f/f* and *Alb-Orc2^f/f* male mice. *p<0.05, **p<0.01, two-tailed Student's t-test.

The online version of this article includes the following source data and figure supplement(s) for figure 2:

**Source data 1.** Original Western blot membrane picture corresponding to *Figure 2*, panel C.

*Figure 2 continued on next page*

*Figure 2 continued*

**Source data 2.** Original image for *Figure 2*, panel C.

**Source data 3.** Original Western blot membrane picture corresponding to *Figure 2*, panel C.

**Source data 4.** Original image for *Figure 2*, panel C.

**Figure supplement 1.** Serum liver enzymes in *ORC2 wild-type* (*Orc2^{f/f}*) and *Orc2 KO* (*Alb-Orc2^{f/f}*) mice.

There was some elevation of the circulating liver enzymes in the mice where Alb-Cre is expressed to delete the *Orc2* gene, suggesting that there is some impairment of liver function (***Figure 2—figure supplement 1***).

The livers of mice can grow by proliferation (increasing the number of cells) or by hypertrophy of cells (larger cells with endoreduplicated nuclei). Histological examination of the livers revealed that the *Orc2* deletion was accompanied by the presence of fewer (~50% of WT levels), but larger nuclei and cells (***Figure 2H and I***). The *Orc2* deleted hepatocytes had significantly larger nuclei in both males and females (***Figure 2J and K***).

To determine whether the larger nuclei were generated by endoreduplication, we isolated nuclei from hepatocytes and stained them with the DNA staining dye, DRAQ5, followed by flow cytometry (***Figure 3A***). Both male and female *Alb-Cre, Orc2^{f/f}* hepatocytes showed a decrease in 4 n DNA content and an increase in 16 n DNA content (***Figure 3B-D***). The expression of Cre would also delete a lox-stop-lox element upstream from EYFP and lead to the expression of EYFP. Indeed, when we gated on EYFP positive cells, the 8 n and 16 N DNA content increase was much more evident relative to the EYFP negative cells, while the 2 N DNA-containing nuclei were decreased (***Figure 3E-G***). Thus *Orc2* deletion promotes endo-reduplication, a result similar to what was noted when *Orc1* was deleted in mouse livers (***Okano-Uchida et al., 2018***).

## DNA replication of *Alb-Cre^{+/-}-Orc2^{f/f}* hepatocytes in vitro

Primary hepatocytes isolated from 8 to 10-wk-old mouse liver can replicate their DNA and proliferate for a limited time in vitro. We isolated such hepatocytes from *Alb-cre^{-/-}* (WT) and *Alb-cre^{+/-}*, both from *Orc2^{f/f}* mice (***Figure 4A***). Genotyping reveals that over 90% of the cells from the *Alb-cre^{+/-}* have successfully deleted *Orc2* exons 6 and 7 (***Figure 4B***), while immunoblotting shows that ORC2 protein expression is also significantly decreased in these cells (***Figure 3C***). EdU labeling in vitro for 2 hr, showed that the EYFP positive cells (where Cre recombinase has been active) can incorporate EdU (***Figure 4C***), although the total number of nuclei that incorporate EdU is decreased to about 30% of that seen in the ORC2 wild-type hepatocytes (***Figure 4D***). Thus hepatocytes in culture can continue to replicate DNA in the absence of detectable ORC2 protein.

## DNA replication in *Alb -Orc2^{f/f}* hepatocytes during liver regeneration in vivo

Liver DNA synthesis peaks around 36–48 hr, and the liver regenerates to nearly 50% of its original weight within 2–3 d following partial hepatectomy. Such regeneration involves both normal mitotic DNA replication/cell division and endoreduplication accompanied by hypertrophy of the cells. We therefore tested whether deletion of the *Orc2* gene in the hepatocytes adversely affects liver regeneration after partial hepatectomy in 8–14-wk-old mice (***Figure 5A***). Nearly two-thirds of the liver is removed surgically and the mice allowed to recover for 36–48 hr before harvesting the regenerated liver.

Although the liver weight after regeneration was smaller than that of WT livers, the liver:body weight ratio was similar in the livers with *Orc2* deletion and *Orc2* WT (***Figure 5B-D***). The H&E stain of the regenerated liver shows that the liver cells are larger and have larger nuclei in the liver with *Orc2* deletion (***Figure 5E***, similar to ***Figure 2***). In these experiments, both before and after regeneration, the livers with the *Orc2* deletion have ~50% nuclei compared to the wild-type livers (***Figure 5F***). Finally, labeling the livers by injection of EdU in the mice 3–4 hr before harvesting, shows that the EYFP positive cells (indicating activity of Cre recombinase) were proficient in synthesizing DNA and incorporating EdU (***Figure 5G and H***). Nearly 100% of the hepatocytes were positive for EYFP, suggesting that the Cre recombinase was active in the vast majority of the hepatocytes (***Figure 5G***), and consistent with the complete depletion of ORC2 protein from

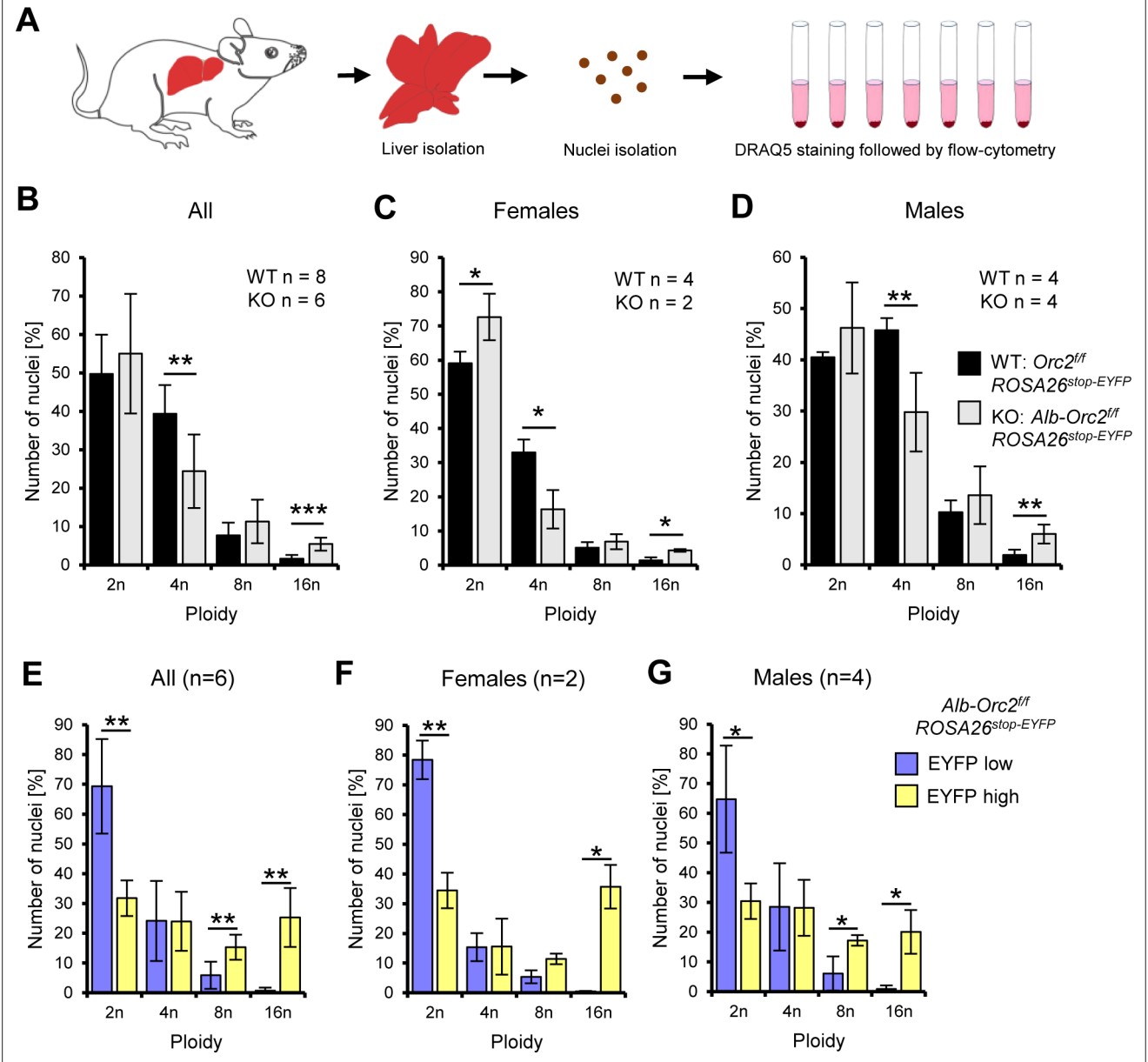

**Figure 3.** Endoreduplication in the ORC2 liver conditional knock-out animals. (**A**) Experimental design. (**B–D**) Quantification of nuclei ploidy on 10,000 nuclei from the livers of *Orc2ᶠ/ᶠ ROSA26ˢᵗᵒᵖ⁻ᴱʸᶠᴾ* and *Alb-Orc2ᶠ/ᶠ ROSA26ˢᵗᵒᵖ⁻ᴱʸᶠᴾ* animals. (**E–G**) Quantification of nuclei ploidy for EYFP low (includes negative) and high (positive) primary liver cells. *p<0.05, **p<0.01, ***p<0.001, two-tailed Student's t-test.

the hepatocytes of all mice with that genotype (*Figure 3C*). Thus the 30% of the liver cells that were incorporating EdU in the Cre expressing livers (*Figure 5H*) were doing so in the absence of ORC2 protein. Also, we noted several pairs of EdU positive nuclei (marked by arrows in *Figure 5G*) whose relative positions suggest that they are sisters born from the same mitosis, suggesting that some EdU positive cells can go through mitosis in the EYFP positive cells. Finally, the nuclei were significantly larger in the Cre active *Orc2ᶠ/ᶠ* hepatocytes than in the hepatocytes without Cre, both before and after regeneration (*Figure 5I*).

We did not explore why the EYFP protein is mostly nuclear in hepatocytes in culture (*Figure 4C*) and mostly cytoplasmic in hepatocytes in the liver tissue (*Figure 5G*), but speculate that differences in signaling pathways or fixation techniques between the two conditions contribute to this difference.

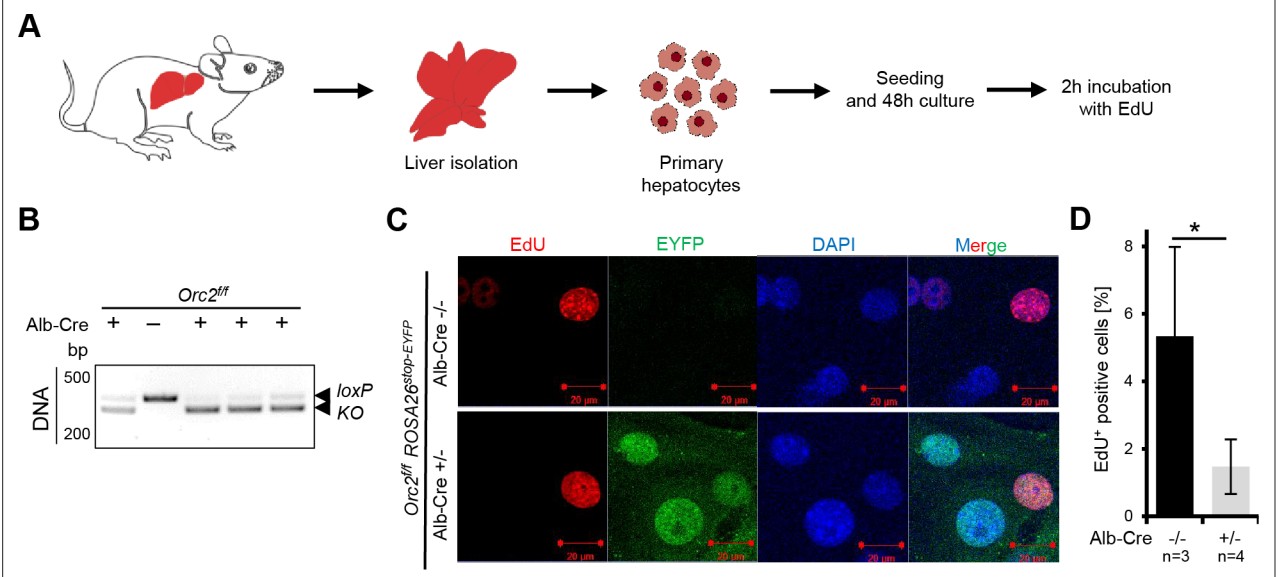

**Figure 4.** *Orc2* KO primary hepatocytes are viable and can incorporate EdU in vitro. (**A**) Experimental design. (**B**) Genotyping and western blotting of hepatocytes. (**C**) Representative picture of EdU, EYFP and DAPI staining on the *Orc2* WT (*Orc2^f/f*) and KO (*Orc2^f/f Alb-Cre*) primary hepatocytes. (**D**) The percentage of EdU positive nuclei from *Orc2* WT or *Orc2* KO primary hepatocytes. *p < 0.05, two-tailed Student's t test.

The online version of this article includes the following source data for figure 4:

**Source data 1.** PDF file containing original DNA gel picture corresponding to *Figure 4*, panel B, indicating the relevant bands and individual animals.

**Source data 2.** Original image for *Figure 4*, panel B.

## Viable mice with endoreduplicated hepatocyte nuclei in *Alb-Orc1^f/f Orc2^f/f* mice

We have reported that conditional deletion of *Orc1* in developing mouse livers still allowed livers to develop and induced premature endoreduplication, suggesting that significant DNA synthesis can occur in liver cells that are genetically deleted of *Orc1*. We bred the *Orc1^f/f* mice with *Orc2^f/f* mice to obtain mice where both *Orc1* and *Orc2* are floxed but *Alb-cre* is not present (*Figure 6A and B*). Even though Cre was not expressed in these livers, for unknown reasons there was a decrease in the percentage of progeny when *Orc2* was floxed in the liver (with or without *Orc1* being floxed), but not when just *Orc1* was floxed (*Figure 6C*). These mice were then bred with *Orc2^f/f Alb-cre^+/+* mice, and the resulting *Orc1^f/+, Orc2^f/f, Alb-cre^+/-* intercrossed to get *Orc1^f/f, Orc2^f/f, Alb-Cre^+/-* mice where the *Orc1* and *Orc2* alleles are conditionally deleted in hepatocytes, and the deletion is accompanied by activation of EYFP expression. Immunoblotting of isolated hepatocytes showed that in four out of four mice expressing the Cre recombinase the ORC1 and ORC2 proteins were decreased significantly (*Figure 6D*). Immunoblotting of some of the other replication proteins showed no decrease in other ORC subunits, CDC6, and two of the MCM2-7 subunits (*Figure 6E*). The ORC3 protein, which did not change when *Orc2* was deleted (*Figure 2D*) was surprisingly elevated in *Figure 6E* when *Orc1* and *Orc2* were both deleted. We do not know why this is the case, and it was not seen consistently in all four animals. Similarly, the 50% decrease of MCM2 and MCM3 that was seen in the absence of ORC2 (*Figure 2D*), disappeared when both *Orc*1 and *Orc*2 are deleted.

The female *Alb-Orc1^f/f Orc2^f/f* mice were significantly smaller in size, with smaller livers and decreased liver to body weight ratio (*Figure 6F, Figure 6—figure supplement 1A, Figure 6—figure supplement 2A*), but the male mice were relatively less affected (*Figure 6F*). We have also observed ~50% lethality of double knock-out female mice within the first month of life, but the remaining 50% survive beyond 4 mo of age (*Figure 6—figure supplement 1C*). H&E staining showed that the double knockout (DKO) male livers have fewer but larger cells with significantly larger nuclei (*Figure 6G and H, Figure 6—figure supplement 2C*). Interestingly, this phenotype is even more marked in the female liver (*Figure 6—figure supplement 1B*). By flow cytometry, the EYFP positive cells as evidence of Cre recombinase activity had distinctly more polyploidization than the EYFP negative cells (*Figure 6I*).

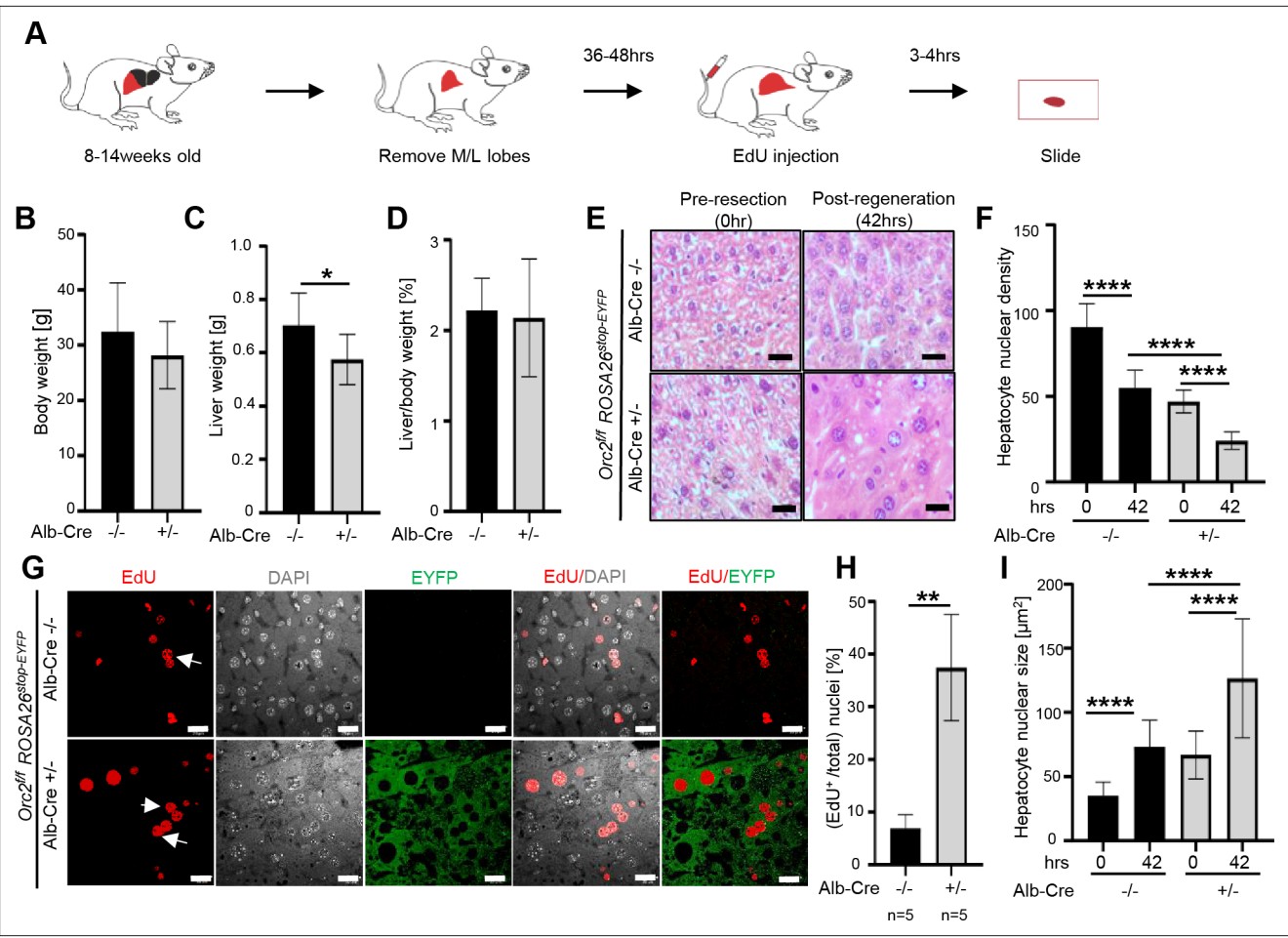

**Figure 5.** The ORC2 mutant livers regenerate after partial hepatectomy. (**A**) Schematic of the experiment. (**B**) Body weight of the *Orc2^f/f ROSA26^stop-EYFP* mice without (-/-) or with *Alb-Cre* (+/-) before partial hepatectomy. (**C**) Liver weight of the mice in B after liver regeneration. (**D**) Regenerated liver to pre-hepatectomy body weight ratio of the mice in B. (**E**) H&E stain of *Orc2^f/f ROSA26^stop-EYFP* livers with intact *Orc2* (*Alb-cre^-/-*, N=3) or *Orc2* knockout (*Alb-cre^+/-*, N=7). Scale bar: 25 µm. (**F**) Quantitation of nuclear counts per field (76,000 µm²). Six images were taken for each liver. 0 hr (pre-resection). 42 hr (post-regeneration). (**G**) EdU incorporation of indicated livers. EYFP marks cells where Cre has been expressed. *Orc2* (*Alb-cre^-/-*, N=5) or *Orc2* knockout (*Alb-cre^+/-*, N=5). Scale bar: 25 µm. (**H**) Percent EdU+ nuclei counted in 1882 and 825 nuclei in the *Cre-* and *Cre+* livers, respectively. (**I**) Nuclear size of indicated livers. 0 hr (pre-resection). 42 hr (post-regeneration). Mean and S.D from about 40–70 nuclei, *p<0.05, ****p<0.0001, unpaired two-tailed Student's t test is used.

Taken together the results reveal that hepatocytes can synthesize DNA by endoreduplication to produce very large nuclei and very large cells so that the liver size is not hugely decreased, even in the absence of two subunits of ORC. Female mice that have a deletion in two subunits of ORC suffer more morbidity and mortality than male mice.

## DNA replication in *Alb-Cre^+/- -Orc1^f/f Orc2^f/f* hepatocytes during liver regeneration in vivo

Finally, partial hepatectomy was performed in the livers of male mice with DKO of *Orc1* and *Orc2*. As can be seen even before partial hepatectomy the body weights were not significantly different from WT mice (*Figure 7A*). Post regeneration, the liver weights and liver/body weight ratios were not decreased in the DKO mice compared to the WT mice (*Figure 7B and C*). H&E sections showed that the hepatocyte nuclei were larger and hepatocyte nuclear density lower in the DKO livers both pre-hepatectomy and in the regenerated livers 36 hr post-hepatectomy (*Figure 7D–F*). EdU labeling for 3–4 hr before harvesting the livers showed that despite the DKO, the EYFP positive cells showed extensive DNA synthesis (*Figure 7G*). Even though nearly 100% of the hepatocytes are EYFP positive (*Figure 7G*) and ORC1 and ORC2 proteins depleted in the hepatocytes (*Figure 6D*), 15% of nuclei

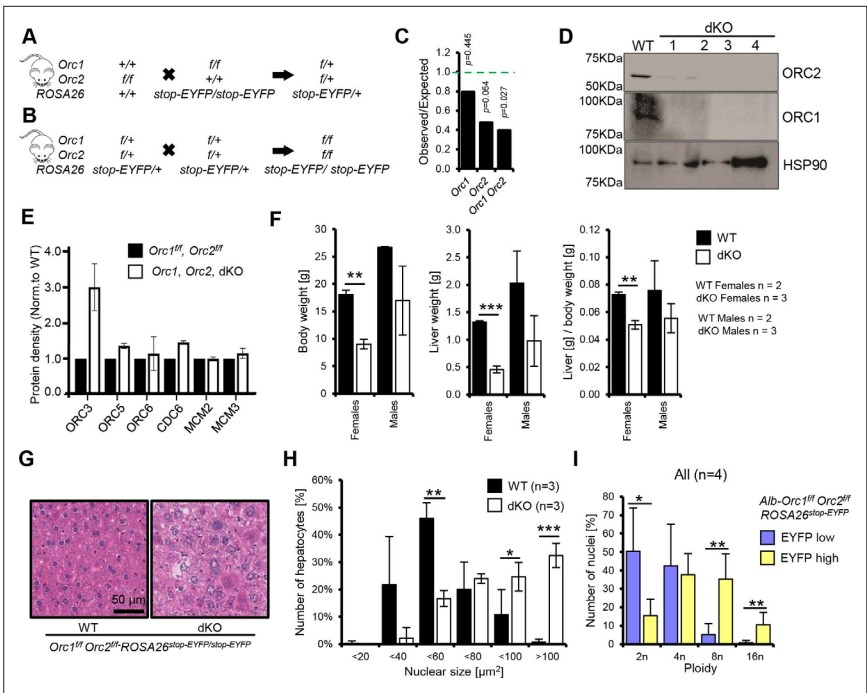

**Figure 6.** Endoreduplication in liver of *Orc1 Orc2 DKO* animals. (**A–B**) Breeding schemes to obtain conditional double flox animals. (**C**) The ratio of observed to expected animals coming from B. *Orc1*=all animals with *Orc1^f/f^ ROSA26^stop-EYFP^*, *Orc2*=all animals with *Orc2^f/f^ ROSA26^stop-EYFP^*, *Orc1 Orc2*=all animals with *Orc1^f/f^ Orc2^f/f^ ROSA26^stop-EYFP^* genotype. This was before the introduction of *Alb-Cre*. (**D**) Immunoblot of hepatocytes from WT (*Orc1^f/f^ Orc2^f/f^*) and DKO (*Orc1^f/f^ Orc2^f/f^ Alb-cre^+/-^*) mice to show that ORC1 and ORC2 are depleted in the DKO cells. (**E**) Quantitation of immunoblots to show that levels of other key initiation protein subunits are not decreased in the DKO mice hepatocytes. (**F**) Average body, liver weight, and their ratio for WT and DKO animals. (**G**) Representative H&E staining of liver tissue from male WT and DKO animals. (**H**) Quantification of hepatocyte nuclear size in the WT and DKO animals. (**I**) Quantification of nuclei ploidy for EYFP low (includes negative) and high (positive) primary liver cells from DKO mice.

The online version of this article includes the following source data and figure supplement(s) for figure 6:

**Source data 1.** PDF file containing original Western blot membrane picture corresponding to *Figure 6*, panel D, indicating the relevant bands, ORC1 protein expression, and individual animals.

**Source data 2.** Original image for *Figure 6* panel D, ORC1 protein expression, and individual animals.

**Source data 3.** PDF file containing original Western blot membrane picture corresponding to *Figure 6*, panel D, indicating the relevant bands, ORC2 protein expression, and individual animals.

**Source data 4.** Original image for *Figure 6* panel D, ORC2 protein expression, and individual animals.

**Source data 5.** PDF file containing original Western blot membrane picture corresponding to *Figure 6*, panel D, indicating the relevant bands, HSP90 protein expression, and individual animals.

**Source data 6.** Original image for *Figure 6* panel D, HSP90 protein expression, and individual animals.

**Figure supplement 1.** *Orc1 Orc2* deletion produces smaller livers with larger nuclei and more mortality in female mice.

*Figure 6 continued on next page*

*Figure 6 continued*

**Figure supplement 2.** From an independent breeding experiment to generate mice with *Orc1*[-/-]; *Orc2*[-/-] and *Orc1*[-/-], *Orc2*[-/-] livers.

stained with EdU (*Figure 7H*), a percentage that was higher in the DKO regenerating livers than in the WT livers, suggesting that the endoreduplication that accompanies liver regeneration can occur even after deletion of two of the six-subunits of ORC.

## Discussion

The six-subunit ORC is important for recruiting MCM2-7, the core of the replicative DNA helicase and is essential for all forms of DNA replication in eukaryotic cells (*Costa and Diffley, 2022*; *Stillman, 2022*; *Hu and Stillman, 2023*). Yet, it has been possible to select cancer cell lines that have mutations in *Orc1*, *Orc2*, or *Orc5* and do not express detectable levels of the proteins and yet load MCM2-7 to the chromatin and replicate their DNA as they proliferate in culture (*Shibata et al., 2016*; *Shibata and Dutta, 2020*). This paradox suggested that there may be an alternative way to load sufficient MCM2-7 on chromatin and support DNA replication, at least in cancer cells in culture. In addition, there have been two instances where *Orc1* has been mutationally removed in *Drosophila* (*Park and Asano, 2008*) and in mouse hepatocytes or placental trophoblasts (*Okano-Uchida et al., 2018*), where extensive DNA replication (particularly endo-reduplication in hepatocytes and trophoblasts) has persisted, suggesting that an alternative way of loading MCM2-7 may be available in certain unique cell-cycles that are different from the normal DNA replication that occurs during normal mitotic growth of diploid cells.

Five of the six-subunits of ORC (ORC1-5) associate to form a ring-shaped structure with interactions between the WH domains and the AAA + like domains of the five subunits (*Bleichert et al., 2015*). In mammalian cells only ORC1 and ORC4 have the intact Walker A and B motifs that are necessary for the molecules to act as AAA + ATPases (*Giordano-Coltart et al., 2005*). Though this has never been demonstrated experimentally, it is theoretically possible that in the absence of ORC1 in flies or in mouse livers, another related AAA + ATPase, intimately involved in MCM2-7 loading, CDC6, can substitute for ORC1 to reconstitute a functional 5 subunit ring-shaped ORC-like structure that executes its function (*Takeda et al., 2005*; *Bell, 2017*). We therefore wished to test whether removal of a second subunit of ORC, ORC2, that does not have obvious ATPase activity and not much homology to CDC6, would still permit endoreduplication in mouse livers. In the cancer cells, we had removed the initiator methionine of the *Orc2* gene (*Shibata et al., 2016*). Although 99.9% of the ORC2 protein disappeared, an N terminally truncated form of ORC2 protein was expressed in the mutant cells at 0.1% of the wild-type level (*Chou et al., 2021*). Thus, in this case, we took the additional precaution to design a mutation such that even if a truncated protein was expressed from an internal methionine downstream from the mutation site, such a protein will be missing a significant part of its AAA + ATPase domain. Anyhow, in Western blots we do not see any evidence of such a truncated protein being expressed in the genetically altered MEFs or hepatocytes. In addition, we wondered whether the simultaneous genetic removal of two subunits of ORC, *Orc1*, and *Orc2*, would successfully obliterate all forms of DNA replication, including endoreduplication.

Our results show that while *Orc2* is genetically essential for viability of early embryos in utero and mouse embryo fibroblasts in vitro, mutational inactivation of *Orc2*, or combined inactivation of *Orc1* and *Orc2*, does not significantly inhibit development of the mouse liver. Adult, viable mice are produced. Albumin expression is activated very early in hepatoblasts in the 7–8 somite stage and the albumin mRNA can be seen in the hepatic primordium emerging from the gut at E9.5 days (*Gualdi et al., 1996*). Single-cell sequencing studies confirm that hepatoblasts expressing albumin are present at E11.0 and that undifferentiated endodermal cells (the precursors to hepatoblasts) are non-existent (*Wang et al., 2020*) [3]H-thymidine incorporation can be measured in the liver as late as days 7–14 post-natally (*Tilghman and Belayew, 1982*), so that many cycles of DNA replication and cell proliferation are expected to occur in hepatoblasts and hepatocytes after the activation of Alb-Cre and subsequent homozygous knockout of *Orc2* (or *Orc1*) in the hepatocytes.

The presence of functional livers in the mutant (but viable) animals, suggested that the homozygous knockout of these genes did not have a profound effect on liver development, as would have

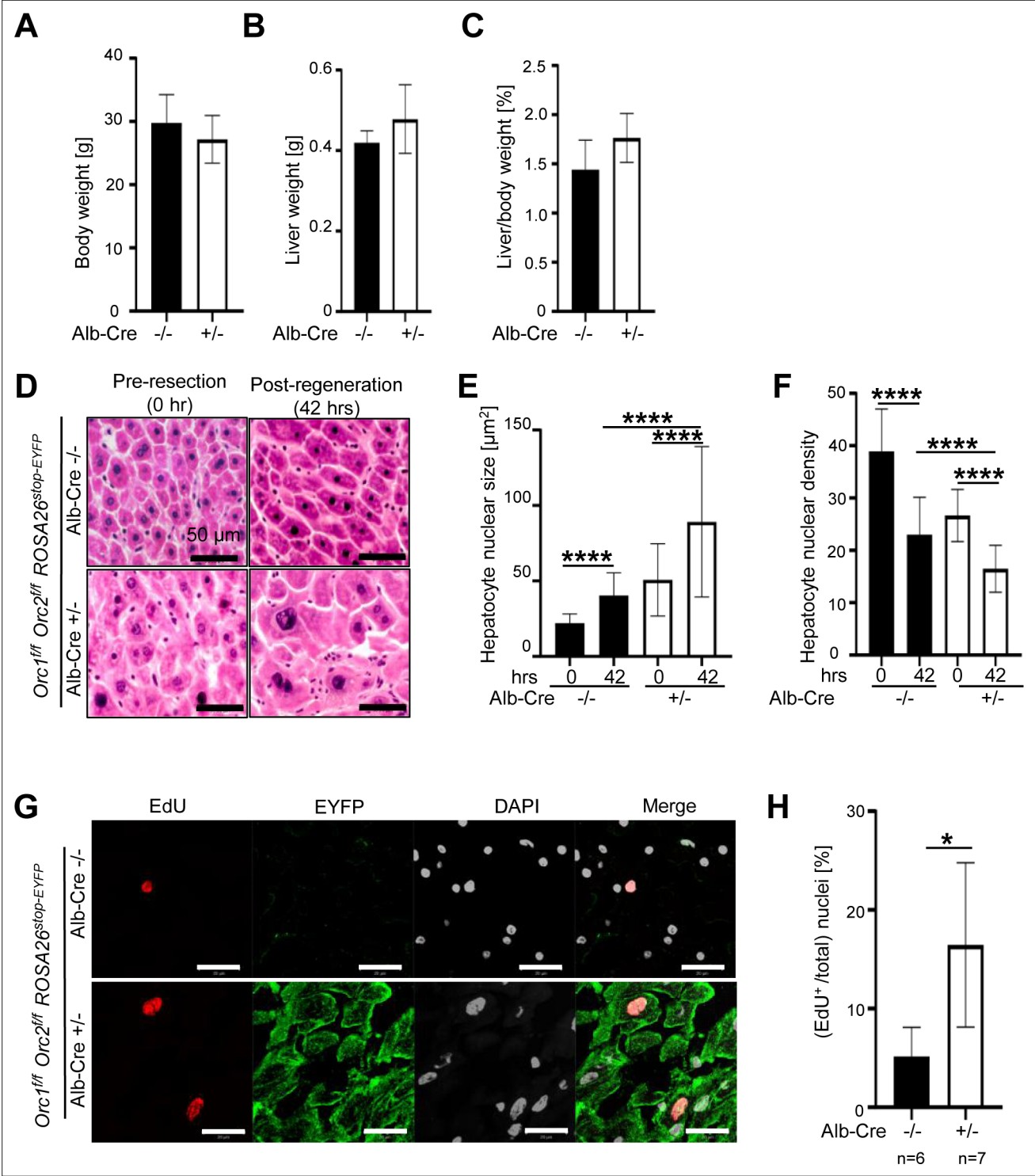

**Figure 7.** Endoreduplication in the liver of the DKO mice during liver regeneration. (**A–C**) Body weight pre-resection, liver weight post-regeneration, and regenerated liver to body weight ratio in mice with indicated genotypes. Black bars: 4 wild-type (WT) males (*Orc1^{f/f} Orc2^{f/f} ROSA26^{stop-EYFP}* mice without *Alb-Cre*). White bars: 6 dKO males (*Orc1^{f/f} Orc2^{f/f} ROSA26^{stop-EYFP}* mice with *Alb-Cre^{+/-}*). No significant difference between the two groups using two-tailed Student t-test. (**D**) H&E stain of WT (N=4) or DKO mice (N=6). Scale bar: 50 μm. (**E**) Quantitation of hepatocyte nuclear size post regeneration. WT: black bars. DKO: white bars. 0 hr (pre-resection). 42 hr (post-regeneration). Five-six images were taken for each liver. About 120–200 nuclei are counted. (**F**) Quantitation of hepatocyte nuclear density post regeneration. WT: black bars. DKO: white bars. 0 hr (pre-resection). 42 hr (post-regeneration). Five-six images were taken for each liver. (**G**) Micrographs of EdU, DAPI and EYFP imaging of livers with indicated genotypes post regeneration. WT (N=6) and DKO mice (N=7). Scale bar: 20 μm. WT in the top row, DKO in the bottom row. (**H**) Quantitation of EdU positive nuclei post

*Figure 7 continued on next page*

*Figure 7 continued*

regeneration. WT: black bar. DKO: white bar. Five-six images were taken for each liver. *p<0.05, ****p<0.0001, unpaired two-tailed Student's t-test were used.

been expected if the cells were as sensitive as MEFs in culture to the loss of the ORC holocomplex. We have been unable to find antibodies that will recognize mouse ORC1 or ORC2 proteins in immunohistochemistry on tissues and so decided to calculate how many cell divisions have to occur *after* the *Orc1* or *Orc2* genes are deleted in the embryonic mouse hepatocytes (see Methods and *Table 2*). The calculations suggest that the *Orc2* deleted livers and male DKO livers underwent at least 18 cell divisions, while the female DKO livers underwent at least 15 cell divisions since E9.5 (*Table 2*, bottom row).

If we allow a generous three cell divisions for the Cre recombinase to definitively excise the Floxed genes (consistent with what we see in MEFs in culture), this means that 15 cell divisions in the *Orc2* deleted livers and male DKO livers occurred *after* the gene(s) was/were deleted. Once a gene is deleted, each cell division decreases the corresponding protein at least by half, so that six cell divisions would dilute the targeted protein to <1.5% of the wild-type levels. Thus even after the ORC2 (or ORC1 and ORC2 in the DKO mice) decreased to <1.5% the WT level, the *Orc2* knockout livers (male or female) and the male DKO livers could execute at least 9 more rounds of replication and cell division. Note the amount of the relevant ORC subunit would continue to decrease by half with each further division. Also, that the levels of ORC are likely to decrease faster if the protein is actively degraded or if more cell divisions are necessary to counter any development-related apoptosis.

This suggests that mitotic DNA replication of diploid cells is not completely eliminated at least in the *Orc2*-/- or male DKO livers after loss of the targeted ORC proteins. Consistent with this, we see EdU incorporation in EYFP + hepatocytes, in vitro, which, however, could be from a terminal endo-reduplication event. More tantalizing is the presence of *paired* EYFP positive EdU-positive nuclei during the regeneration after partial hepatectomy in 8–14-wk-old mice (*Figure 5G*). Such paired nuclei are usually seen with daughter nuclei that have just been born from mitosis, suggesting that there were at least a few cells that underwent complete DNA replication and mitosis even though the hepatocytes had undergone 15 rounds of cell division after deletion of the targeted ORC gene(s).

Could the ORC deletion lead to the immediate loss of hepatoblasts (despite having inherited ORC protein from the endodermal cells) causing undifferentiated endodermal cells to persist and proliferate much longer (without activating Cre) than in normal development? We consider this unlikely, but if true it will be very unexpected, both by suggesting that deletion of ORC immediately leads to the death of the hepatoblasts (despite a healthy reserve of inherited ORC protein) and by suggesting that there is a novel feedback mechanism from the death/ORC depletion of hepatoblasts leading to the persistence and proliferation of undifferentiated endodermal cells.

The female DKO (but not the *Orc2*-/-) livers seem more sensitive to the depletion of both ORC1 and ORC2 since ~12 cell divisions in the female DKO livers occur after allowing three cell division for the targeted genes to be definitively deleted. Thus, after the cells have reached <1.5% of WT levels of ORC2 or ORC1 protein, they can still execute at least 6 more cell divisions. Consistent with this sex-specific difference, the male mice appear to be more tolerant of deletion of *Orc2* gene or both

**Table 2.** Estimate of number of hepatocyte nuclei in adult mice of indicate genotypes and thus, number of hepatocyte nuclear divisions required after E9.5 mouse embryos.

| | WT | Orc2 -/- | Orc1 -/- Orc2 -/- (Females) | Orc1 -/- Orc2 -/- (Males) |
|---|---|---|---|---|
| Liver weight | 100% | 50–70% | 30% | 47% |
| Hepatocyte nuclear density | 1 | 0.5 | 0.1 | 0.66 |
| Total nuclei in liver (normalized to WT) | 100 | 25–37.5 | 3 | 31 |
| Deficit in # cell division | 0 | 1–2 | 5 | 1–2 |
| Number of nuclear divisions since E9.5 | 20 | 18 | 15 | 19 |

*Orc1* and *Orc2* genes, with the livers reaching larger sizes than in the female mice. Plasma testosterone levels reach 50% of adult levels in the first day after birth before declining (to rise again at puberty at 4–5 wk of age), suggesting that enough androgens are present during the proliferative phase of liver development (*Bell, 2018*). Sex-specific changes in gene expression have been noted to begin in the liver by 3 wk postnatally (*Conforto and Waxman, 2012*). Thus, it is possible that androgens directly or indirectly stimulate the division or hypertrophy of hepatocytes to ameliorate the negative effects on liver mass due to the loss of ORC. We have also shown that loss of ORC subunits in cancer cell lines causes significant changes in gene expression due to the role of ORC subunits in epigenetic regulation (*Su et al., 2025*), so another explanation of the more severe phenotype in females could be that the epigenetic effects of ORC subunits are more important in female compared to male livers.

In contrast to diploid DNA replication in hepatocytes, we have shown that the livers with a genetic deletion of *Orc2* (or *Orc1* and *Orc2*) undergo endo-reduplication more easily during normal development. Such DNA synthesis is also clearly seen during liver regeneration after partial hepatectomy in the *Orc2^-/-^* or *Orc1^-/-^, Orc2^-/-^* livers. Endoreduplication differs from normal replication in that although the cycles of DNA replication (S phase) are separated by a G1 like phase (G phase) they are not separated by an intervening mitosis and cytokinesis, and is commonly seen during normal development in plants and animals (*Shu et al., 2018*). However, all evidence suggests that the biochemical mechanism of initiating DNA replication is the same in normal mitotic replication and endoreduplication. For example, cyclin E/cdk kinase activity and MCM2-7 protein association with the chromatin oscillate between G phase and S phase during *Drosophila* endoreduplication (*Lilly and Duronio, 2005*; *Su and O'Farrell, 1998*). Similarly, in the mouse, cyclin E and CDC6 promote, while geminin inhibits endoreduplication (*Welch, 1992*; *Bermejo et al., 2002*; *Gonzalez et al., 2006*; *de Renty et al., 2014*). Thus, the persistence, and in fact acceleration, of endoreduplication in the mouse livers in the absence of ORC2 and/or ORC1 suggests that there must be an alternate way to recruit sufficient MCM2-7 to the chromatin to support at least the three cycles of endoreduplication required to produce a 16 N nucleus from a 2 N embryonic nucleus during normal development, and to support about two cycles of endoreduplication during liver regeneration.

The acceleration and persistence of endoreduplication even when both *Orc1* and *Orc2* are genetically inactivated, makes it unlikely that a make-shift ORC-like complex is being assembled for loading the MCM2-7 proteins when two out of the five subunits in the ring are missing. Our results are virtually identical to what was observed when we conditionally deleted *Orc1* (*Okano-Uchida et al., 2018*). There too, endoreduplication in the liver cells not only persisted but was induced prematurely during development. One possible explanation of the greater endoreduplication in both our papers is that mitosis may be arrested earlier in development by G2 DNA damage checkpoints activated by incomplete licensing and replication of the genome in the absence of ORC. As a result, endoreduplication cycles could begin earlier in development resulting in greater endoreduplication.

Is it possible that incomplete deletion of *Orc2* or *Orc1* genes in the hepatocytes allows enough liver cells to still carry an *Orc2* or *Orc1* gene to support DNA replication? This is very unlikely, because the in vitro DNA replication experiment with hepatocytes shows that although the ORC2 protein is virtually undetectable in immunoblots and there is evidence of Cre activity in 100% of the hepatocytes, the number of nuclei incorporating EdU in culture are decreased to only 30% the wild-type level. Furthermore, in the partial hepatectomy experiments, nearly 100% of the hepatocytes are positive for EYFP and no ORC2 (or ORC1 protein in the DKO) is detectable in immunoblots of isolated hepatocytes, suggesting high penetrance in the expression of the Cre recombinase in the hepatocytes. Yet the liver succeeds in endoreduplicating to reach nearly normal liver size and EdU incorporation is seen in 35% (*Figure 5H*) and 15% (*Figure 7G*) of hepatocytes from the *Orc2* KO and DKO mice, respectively.

Although the liver size in the *Alb-Cre, Orc2^f/f^* mice is not significantly decreased relative to body weight, the liver function tests suggest some impairment of liver function. We cannot yet attribute this deficit of liver function to the decrease in number of cells, or to excess endoreduplication, because ORC subunits are also known to be important for epigenetic control of gene expression (*Vermeulen et al., 2010*). Future experiments will determine whether the liver pathology could be secondary to epigenetic dysregulation of genes important for liver function. The double knockout female mice are sicker, and here some of the explanation may lie in the smaller liver, but here again, epigenetic dysregulation in the absence of ORC cannot be ruled out as a potential cause for the morbidity.

Interestingly, the deletion of *Orc2* or of *Orc1* + *Orc2*, in the hepatocytes does not *consistently* increase or decrease any of the other ORC subunits or the examined proteins downstream of ORC like CDC6 or two of the subunits of MCM2-7. Thus, a change in any of these proteins is unlikely to be the explanation for how the hepatocytes license enough origins to support replication during development and regeneration in the absence of the ORC holocomplex. The hepatocyte studies say that the cancer cells are not unique in their ability to bypass the requirement of two ORC subunits. Even primary cells can sometimes load enough MCM2-7 to replicate DNA in the absence of detectable amounts of the ORC holocomplex. It is also worth noting that in cancer cells in culture, we are seeing a near normal level of loading of MCM2-7 on chromatin when *Orc1*, *Orc2*, or *Orc5* genes are deleted, and that 60% of the origins of replication remain at the same sites as in WT cells (*Shibata et al., 2024*). Although it is impossible to rule out that a very small amount of ORC protein somehow persists in the hepatocytes (or the cancer cell lines) with these mutations and that this is sufficient to facilitate loading of enough MCM2-7 to support DNA synthesis in vivo or in vitro, this is becoming progressively unlikely.

## Methods

### Mice

Work involving mice adhered to the guidelines of the Institutional Animal Care and Use Committees (IACUC) at the University of Virginia (protocol number 4198), the University of Alabama at Birmingham (protocol number 22335), the Ohio State University and the Medical University of South Carolina. *Orc1* ^f/f^ *ROSA26*^stop-EYFP^ animals were previously reported (*Okano-Uchida et al., 2018*). *Orc2* ^f/f^ mice were generated by Cyagen Biosciences Inc Exons 6–7 (amino acids L111-V150) was selected as conditional knockout region (cKO). Mouse genomic fragments containing homology arms (HAs) and cKO region were amplified from BAC clone by using high fidelity Taq DNA polymerase and were sequentially assembled into a targeting vector together with recombination sites and selection marker, Neo cassette, flanked by SDA (self-deletion anchor) sites. The linearized vector was electroporated into C57BL/6 ES cells that were subject to G418 selection (200 μg/mL) after 24 hr. 188 G418 resistant clones were picked and amplified in 96-well plates. The PCR screening identified 29 potential targeted clones, from among which 12 were expanded and further characterized by Southern blot analysis. Eleven of the twelve expanded clones were confirmed to be correctly targeted. Targeted ES cell clone N-1F10 was injected into C57BL/6 albino embryos, which were then re-implanted into CD-1 pseudopregnant females. Founder animals were identified by their coat color, their germline transmission was confirmed by breeding with C57BL/6 females and subsequent genotyping of the offspring. Three male and five female heterozygous targeted mice were generated from clone N-1F10. Floxed *Orc2* mice were crossed to the *Orc1* ^f/f^ *ROSA26*^stop-EYFP^ animals to generate *Orc2* ^f/f^ *ROSA26*^stop-EYFP^ and *Orc1* ^f/f^ *Orc2* ^f/f^ *ROSA26*^stop-EYFP^ strains for usage of EYFP expression as a reporter for Cre recombinase activity and *Orc2* or *Orc1* and *Orc2* deletion. Those were further bred into tissue-specific *Sox2*-Cre (*Hayashi et al., 2002*) or *Alb*-Cre mice to obtain *Orc2* or *Orc1* and *Orc2* knockout in all cells of the inner cell mass following implantation or in the liver, respectively. We used two independent *Alb*-Cre lines to introduce the gene. At UVA/UAB we used the *Alb*-Cre mice from *Postic et al., 1999*. The *Alb*-Cre mice used at the University of Wisconsin are from *Schuler et al., 2004*. All the mice used in this study were maintained in a mixed background (C57BL/6x129 x FVB/N).

### PCR

Genomic DNA from ear punches was isolated using Quick Extract DNA Extraction Solution (Lucigen., Cat# QE09050). All genotyping PCRs were carried out using MyTaq Red Mix (Bioline, Cat# BIO-25043) according to the manufacturer's instructions. *Orc*1 genotyping was carried out with primers F1 forward (common to both alleles; 5'-GCTGCTTCAGTGTGGCAATA-3'), R1 reverse (specific for the WT allele; 5'-CTCCAATTGTTCCCCAGCTA-3'), and R2 reverse (specific for deleted allele, 5'-CACCTGTCACTGGACCACAC-3'). The PCR parameters were 95 °C for 30 s, 45x (95 °C for 15 s, 60 °C for 20 s, 72°for 60 s), 72 °C for 5 min. PCR product was run on 2% agarose gel and WT band was detected at 439 bp, transgenic band at 310 bp, and deleted at 677 bp. *Orc*2 genotyping was carried out with primers F1 forward (common to both alleles; 5'- GAGGTTGTGGCTGTAATATACGTGATC –3'), and R1 reverse (common to both alleles; 5'- CTGAGCCATCTAACTCCTTCCTAGC –3'), or F2 forward (common to

both alleles; 5′- TGGGTAGGTTCATTCCAGTTTAGCC –3′), and R2 reverse (common to both alleles; 5′-ACCTTGGTATTGGACGTCTCTATTC –3′). The PCR parameters were 95 °C for 30 s, 35x (95 °C for 15 s, 55 °C for 20 s, 72°for 60 s), 72 °C for 5 min. To detect constitutive KO allele combination of F1, F2, and R2 was used. PCR product was run on 2% agarose gel and following bands were detected: for F1 + R1 – WT at 342 bp, and transgenic band at 398 bp; for F2 +R2 WT at 254 bp, and transgenic band at 367 bp; for F1+F2+R2 WT at 254 bp, transgenic band at 367 bp, and deleted at 306 bp. ROSA26 genotyping was carried out with primers F1 forward (common to both alleles; 5′-AAAGTCGCTCTG AGTTGTTAT-3′), R1 reverse (specific for the WT allele; 5′- GGAGCGGGAGAAATGGATAT-3′), and R2 reverse (specific for deleted allele, 5′- GCGAAGAGTTTGTCCTCAACC-3′). The PCR parameters were 95 °C for 30 s, 45x (95 °C for 15 s, 60 °C for 20 s, 72°for 60 s), 72 °C for 5 min. PCR product was run on 2% agarose gel and WT band was detected at 650 bp, and transgenic band at 340 bp. Sox2-Cre genotyping was carried out with primers F1 forward (common to both alleles; 5′- ATGCTTCTGTCC GTTTGCCG –3′) and R1 reverse (common to both alleles; 5′- CCTGTTTTGCACGTTCACCG –3′), with $Orc1$ primers F1 and R1 for an internal control. The PCR parameters were 94 °C for 3 min, 38x (94 °C for 30 s, 60 °C for 30 s, 72 °C for 40 s), 72 °C for 3 min. PCR product was run on 3% agarose gel and the transgenic band was detected at 875 bp with an internal control band at 439 bp. Alb-cre geno-typing was carried out with primers Alb-cre-20239-F (specific for the WT allele; 5′-TGCAAACATCAC ATGCACAC-3′), Alb-cre-olMR5374-F (specific for the transgenic allele; 5′-GAAGCAGAAGCTTAGG AAGATGG-3′), and Alb-cre-20240-R (common to both alleles; 5′-TTGGCCCCTTACCATAACTG-3′). The PCR parameters were 95 °C for 30 s, 35x (95 °C for 15 s, 55 °C for 20 s, 72°for 60 s), 72 °C for 5 min. PCR product was run on 4% agarose gel and WT band was detected at 351 bp, and transgenic band at 390 bp. To establish sex of embryos isolated for MEFs, SRY genotyping was carried out with chromosome Y specific forward (5′-TTGTCTAGAGAGCATGGAGGGCCATGT-3′) and reverse primers (5′-CTCCTCTGTGACACTTTAGCCCTCCGA-3′). The PCR parameters were 95 °C for 30 s, 35x (95 °C for 15 s, 55 °C for 20 s, 72°C for 60 s), 72 °C for 5 min. PCR product was run on 2% agarose gel and Y-chromosome positive band was detected at 270 bp.

## MEF isolation and culture

MEFs were isolated from E12.5 $Orc2^{+/+}$ or $Orc2^{f/f}$ embryos, transformed with SV40 large T antigen, and infected with adenovirus Cre-eGFP (#VVC-U of Iowa-1174, University of Iowa). The infected MEFs were cultured for indicated days in DMEM with 10% FBS medium. To measure the cell proliferation, 24 hr. after the Adenovirus Cre-eGFP transduction, SV40 transformed MEF cells were plated in 96 well plates.

The cell viability was measured every 24 hrs using CellTiter 96 Non-Radioactive Cell Prolifera-tion Assay (Promega, #G4100) according to the manufacturer's instructions. All experiments were conducted in triplicate and absorbance relative to that on day 1 was expressed.

## Liver isolation

Control ($Orc2^{f/f}$ $ROSA26^{stop-EYFP}$ or $Orc1^{f/f}$ $Orc2^{f/f}$ $ROSA26^{stop-EYFP}$) and experimental animals ($Alb-Orc2^{f/f}$ $ROSA26^{stop-EYFP}$ or $Alb-Orc1^{f/f}$ $Orc2^{f/f}$ $ROSA26^{stop-EYFP}$) were euthanized using $CO_2$. Blood was collected was further metabolic studies. Animals were perfused with prewarmed (39 °C) Hank's buffered salt solution (HBSS) containing EDTA, $MgCl_2$, and HEPES. The livers were dissected and weighted. Each liver was divided for the following experiments: ploidy analysis (right lobe; fresh processing), EYFP flow cytometry (median lobe; fresh processing), immunoblotting and histology (left lobe; half for $LN_2$ flash freeze and second half into 10% formalin), and genotyping (caudate lobe; $LN_2$ flash freeze).

## Immunoblotting

SV40 transformed MEF with or without adenovirus Cre-eGFP infection were directly lysed in 2 x Laemmli Sample Buffer and sonicated. Liver was lysed in modified RIPA buffer (150 mM Sodium Chlo-ride, 50 mM Tris-HCl, pH 7.4, 1 mM EDTA, 1 mM PMSF, 1% Triton X-100, 1% Sodium Deoxycholic Acid, 0.1% SDS), sonicated, and lysate was clarified by centrifugation. Mouse ORC2 antibody was raised against His tagged full length of ORC2 recombinant protein in Rabbit (Pacific Immunology). The antibodies used in this study are listed: ORC1 (Santa Cruz; sc-28741); ORC3 (Santa Cruz; sc-374231); ORC5 (Boster Biological technology; A03676-1); ORC6 (Santa Cruz; sc-390490); CDC6 (Santa Cruz; sc-9964); MCM2 (Abcam; ab4461); MCM3 (Santa Cruz; sc-9850).

## Histology and analysis

All formalin-fixed paraffin-embedded (FFPE) sections and H&E staining were performed by Research Histology Cores at UVA and UAB. Nuclei size was measured using ImageJ 1.50i (Java 1.6.0_24) (56) (*Schneider et al., 2012*). The number of analyzed animals is annotated at each figure. The statistical method used for comparison between experimental groups was a two-tailed homoscedastic Student's t-test. Statistical significance was expressed as a p-value. We captured images of 5–10 fields per liver and measured nuclear size relative to scale bar and nuclear density by counting the number of nuclei per field (at a fixed scale to compare between mice). Hepatocyte nuclei can be easily distinguished from stromal nuclei by their roundness and relative de-condensation.

## Metabolic measurements

Blood was collected from control *Orc2$^{f/f}$ ROSA26$^{stop-EYFP}$* and experimental *Alb-Orc2$^{f/f}$ ROSA26$^{stop-EYFP}$* animals and centrifuged at 1000 g for 10 min at 4 °C. To measure Alanine Transaminase (ALT) and Aspartame Aminotransferase (AST) activities EnzyChromTM Alanine Transaminase Assay Kit (BioAssay Systems, Cat# EALT-100) and Liquid AST (SGOT) Reagent Set (Pointe Scientific, Cat# A7561150) were used respectively according to manufacturers' instructions. The number of analyzed animals is annotated at the figure. The statistical method used for comparison between experimental groups was a two-tailed homoscedastic Student's t-test. Statistical significance was expressed as a p-value.

## Isolation of hepatocytes

Hepatocytes were isolated according to STAR protocols (*Charni-Natan and Goldstein, 2020*). In brief, the liver perfused with perfusion Buffer for 10 min followed by digestion buffer for 10 min was dissected out and hepatocytes were released into plating medium containing dish. A single-cell suspension was obtained by filtering through a 100 µm cell strainer. Percoll centrifugation (Cytiva #17089102) was performed to remove dead cells. Isolated live Hepatocyte were suspended in William's medium (WEM, GIBCO A1217601) with Plating Supplement (GIBCO #CM3000) and plated on collagen-coated cover glass (5×10^5 cells/6-well). After 3 hr, the medium was exchanged for maintenance medium (William's E Medium (WEM, GIBCO A1217601) supplemented with GIBCO #CM4000).

## Ploidy analysis

Nuclei Isolation Medium (NIM; 250 mM Sucrose, 25 mM KCl, 5 mM MgCl$_2$, 10 mM Tris-Cl, 1 mM DTT, 1 x Protease inhibitor) with 2% paraformaldehyde and 0.1% Triton X-100 was added to the liver pieces that were subsequently homogenized with Pestle A Dounce homogenizer (25 x times). After all samples were processed, they were centrifuged at 1000 g for 10 min at 4 °C. Pellet was resuspended in NIM and equal volume of 50% iodixanol was added. The mixed sample was carefully layered on the top of 29% Iodixanol solution in ultracentrifuge tube and spun at 10,300 rpm for 10 min at 4 °C in ultracentrifuge. The nuclei pellet was resuspended in FACS buffer (1 x PBS, 2.5% (v/v) BSA, 2 mM EDTA, 2 mM NaN$_3$) with 100 µg/mL RNase A. DRAQ5-stained liver nuclei samples were processed using Attune NxT flow cytometer (Life Technologies). Flow cytometry data from 10,000 nuclei were analyzed with FCS express software. The bottom 40% of nuclei on the EYFP axis in the FACS profiles was considered as low EYFP and the top 20% as high EYFP. The EYFP low nuclei are mostly from non-hepatocytes (Kupffer cells, endothelial cells, bile duct cells, contaminating blood cells) and some hepatocytes that have not yet expressed sufficient levels of EYFP and they all serve as a control population. The EYFP high nuclei are exclusively from hepatocytes that have undergone the Cre-mediated recombination a sufficient time back to express high levels of EYFP and are the experimental population with ORC subunit deletion. The statistical method used for comparison between experimental groups was a two-tailed homoscedastic Student's t-test. Statistical significance was expressed as a p-value.

## EdU incorporation in vitro

Two days after plating, Hepatocyte was labeled with 20 µM of EdU (Lumiprobe #10540) for 2 hrs and fixed with 4% paraformaldehyde for 10 min followed by permeabilized with 0.25% Tritonx-100 for 5 min. The fixed cells were incubated with label mix [8 µM Sulfo-Cy3-Azide (Lumiprobe #B1330), 2 mM CuSO4*5H2O, 113 mM Ascorbic Acid] for 30 min. Anti-GFP antibody (Abcam #ab6556) was used to detect EYFP signal after the EdU staining.

## Partial hepatectomy

The partial hepatectomy experiment was conducted following a standardized protocol (*Mitchell and Willenbring, 2008*). Briefly, 8–14 wk-old mice were utilized. After inducing anesthesia with 2% isoflurane and maintaining at 0.2%, the mice were subcutaneously injected with Buprenorphine-SR (0.6 mg/kg) and Carprofen (5 mg/kg). The abdominal wall was shaved and prepared aseptically. A 3 cm long transverse incision was made to expose the xiphoid. The left and middle lobes were tied using 4–0 thread and cut. The incision site was double-checked for bleeding following washed with 0.9% sodium chloride and sutured. Mice were placed on a warm pad for recovery. To detect DNA synthesis, 0.2 mg EDU (Lumiprobe #10540) was injected via the tail vein 3–4 hr before sacrifice. At 36–48 hr post-surgery, samples from the regenerating right liver were collected and weighed. For IHC and IF staining, samples were fixed with 4% formaldehyde, dehydrated, cleared, embedded in paraffin, and sectioned at 5–8 µm thickness. Tissue sections were deparaffinized and rehydrated, and heat antigen retrieval methods were applied in 10 mM sodium citrate buffer (pH = 6.0). Permeabilization with 0.25% Triton X-100 followed EDU incorporation by Click-in reaction for 40 min at room temperature, protected from light. After washing and blocking, an anti-GFP antibody (Abcam, cat# ab6556, 1:500 dilution) and Alexa-488 fluorescent-conjugated secondary antibody (A11029; Life Technologies) were used to detect EYFP signaling. Finally, nuclei were stained with DAPI, and images were captured using Zeiss Confocal microscope and processed using ImageJ and GraphPad software. To improve the EYFP signal, we used frozen sections for the liver regeneration experiments in the *Orc*1, *Orc*2 double knockout mice in *Figure 7*.

## Estimate of nuclear and cell division in hepatocytes during normal development

To estimate the minimal number of cell divisions occurring in the hepatoblasts after the appearance of albumin-driven Cre, we first estimated how many hepatocyte nuclei populate an adult liver. Given that there are 125 million liver cells/gram of tissue (*Sohlenius-Sternbeck, 2006*), we estimate that there are 162.5 (female) to 218 (male) million hepatocytes for the 1.3 gram (female) or 1.75 gram (male) livers (all estimates of liver weight and liver nuclear density are from *Figures 5 and 6*, and *Figure 6—figure supplement 1*).

We next estimated the number of albumin-positive cells early in embryogeneis, from published scRNAseq results that suggest that ~1% of the embryonic cells are of hepatocyte lineage when all embryonic cells are harvested from E9.5-E13.5 embryos (*Cao et al., 2019*). Given that there are ~200,000 cells in E9.5 embryos (*Cao et al., 2019*), we, therefore, estimate that there are 2000 albumin positive cells at the stage when albumin mRNA expression is readily detected. Thus, normal development of the liver requires at least 20 rounds of diploid cell division from the 2000 hepatoblasts/hepatocytes seen in E9.5 embryos to produce the ~200 million hepatocyte nuclei in the adult liver (*Table 2*, bottom row).

From the relative weights of the adult livers (row 1) and the relative densities of the hepatocyte nuclei (row 2) we can estimate the number of hepatocytes in the mutant livers relative to wild-type livers (row 3). The relative deficit of hepatocytes in the mutant livers allows us to estimate how many fewer cell divisions the hepatocytes underwent in the mutant livers relative to the WT livers during development (row 4). Since WT hepatocytes undergo ~20 duplications, we thus estimate that the *Orc2-/-* hepatocytes and the male DKO hepatocytes undergo at least 18 divisions from the 2000 hepatoblast-stage seen in E9.5 embryo, while the female DKO hepatocyte undergo at least 15 divisions (*Table 2*, bottom row). This underestimates the number of divisions because we do not take into account any apoptosis that may be occurring or any endoreduplication cycles.

## Acknowledgements

This work was supported by a grant from the NIH (R01 CA60499 to AD), Wagner fellowship from the University of Virginia (to RKP), and the F99/K00 NCI Predoctoral to Postdoctoral Fellow Transition Award (F99/K00CA253732 to RKP). This publication was also supported in part by the Medical College of Wisconsin Cancer Center Shared Resources. We thank the following funders for grant support: Advancing a Healthier Wisconsin Endowment (GL, TU, AT), and the Dr. Glenn R and Nancy A Linnerson Endowed Fund (GL). We thank the University of Virginia (RRid:SCR_017829) and UAB

Flow Cytometry Core Facilities and Research Histology Core Facilities, that were partially supported by the NCI Grants (P30-CA044579, P30-CA013148), and thank Dr. Ulrike Lorenz from the University of Virginia for the Alb-Cre breeder, and Kody Park for help with genotyping.

## Additional information

### Funding

| Funder | Grant reference number | Author |
|---|---|---|
| National Institutes of Health | R01 CA60499 | Anindya Dutta |
| University of Virginia | Wagner fellowship | Róża K Przanowska |
| National Cancer Institute | F99/K00CA253732 | Róża K Przanowska |
| Medical College of Wisconsin | Advancing a Healthier Wisconsin Endowment | Gustavo Leone Takayuki-Okano Uchida Anthony Trimboli |
| Medical College of Wisconsin | the Dr. Glenn R. and Nancy A. Linnerson Endowed Fund | Gustavo Leone |

The funders had no role in study design, data collection and interpretation, or the decision to submit the work for publication.

### Author contributions

Róża K Przanowska, Data curation, Funding acquisition, Investigation, Methodology, Writing – original draft, Writing – review and editing, Conceptualization, Formal analysis, Validation, Visualization; Yuechuan Chen, Formal analysis, Investigation, Visualization, Methodology, Writing – original draft, Writing – review and editing; Takayuki-Okano Uchida, Etsuko Shibata, Xiaoxiao Hao, Formal analysis, Visualization, Methodology; Isaac Segura Rueda, Kate Jensen, Anthony Trimboli, Yoshiyuki Shibata, Methodology; Piotr Przanowski, Methodology, Data curation; Gustavo Leone, Supervision; Anindya Dutta, Supervision, Funding acquisition, Writing – original draft, Project administration, Writing – review and editing, Conceptualization, Resources

### Author ORCIDs

Róża K Przanowska ⓘ https://orcid.org/0000-0003-1278-6242
Yuechuan Chen ⓘ http://orcid.org/0000-0001-6856-1582
Xiaoxiao Hao ⓘ http://orcid.org/0000-0003-4404-5246
Piotr Przanowski ⓘ http://orcid.org/0000-0002-9191-4769
Anindya Dutta ⓘ https://orcid.org/0000-0002-4319-0073

### Ethics

This study was performed in strict accordance with the recommendations in the Guide for the Care and Use of Laboratory Animals of the National Cancer Institute/NIH/DHHS. All of the animals were handled according to approved institutional animal care and use committee (IACUC) protocols (APN: IACUC-22335) of the University of Alabama at Birmingham (UAB). The Committee on the Ethics of Animal Experiments of the University of Alabama at Birmingham (UAB) approved the protocol. All surgery was performed under anesthesia, and every effort was made to minimize suffering.

Reviewer #1 (Public review): https://doi.org/10.7554/eLife.102915.3.sa1
Reviewer #2 (Public review): https://doi.org/10.7554/eLife.102915.3.sa2
Reviewer #3 (Public review): https://doi.org/10.7554/eLife.102915.3.sa3
Author response https://doi.org/10.7554/eLife.102915.3.sa4

## Additional files

### Supplementary files
MDAR checklist

Source data 1. source data for all figures.

### Data availability
All data generated or analysed during this study are included in the manuscript and supporting files.

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
