## [Editor Report · eLife Assessment]

This **valuable** descriptive manuscript builds on prior research showing that the elimination of Origin Recognition Complex (ORC) subunits does not halt DNA replication. The authors obtain **solid** data using various methods to genetically remove one or two ORC subunits from specific tissues and still observe replication. The replication appears to be primarily endoreduplication, indicating that ORC-independent replication may promote genome reduplication without mitosis. The mechanism behind this ORC-independent replication remains to be elucidated. The study and mutants described herein lay the groundwork for future research to explore how cells compensate for the absence of ORC and to develop functional approaches to investigate this process. The reviewers suggested the observations could be supported by additional experiments. This work will be of interest to those studying genome duplication and replication.

---

## [Referee Report · Reviewer #1 (Public review)]

The origin recognition complex (ORC) is an essential loading factor for the replicative Mcm2-7 helicase complex. Despite ORC's critical role in DNA replication, there have been instances where the loss of specific ORC subunits has still seemingly supported DNA replication in cancer cells, endocycling hepatocytes, and *Drosophila* polyploid cells. Critically, all tested ORC subunits are essential for development and proliferation in normal cells. This presents a challenge, as conditional knockouts need to be generated, and a skeptic can always claim that there were limiting but sufficient ORC levels for helicase loading and replication in polyploid or transformed cells. That being said, the authors have consistently pushed the system to demonstrate replication in the absence or extreme depletion of ORC subunits.

Here, the authors generate conditional ORC2 mutants to counter a potential argument with prior conditional ORC1 mutants that Cdc6 may substitute for ORC1 function based on homology. They also generate a double ORC1 and ORC2 mutant, which is still capable of DNA replication in polyploid hepatocytes. While this manuscript provides significantly more support for the ability of select cells to replicate in the absence or near absence of select ORC subunits, it does not shed light on a potential mechanism.

The strengths of this manuscript are the mouse genetics and the generation of conditional alleles of ORC2 and the rigorous assessment of phenotypes resulting from limiting amounts of specific ORC subunits. It also builds on prior work with ORC1 to rule out Cdc6 complementing the loss of ORC1.

The weakness is that it is a very hard task to resolve the fundamental question of how much ORC is enough for replication in cancer cells or hepatocytes. Clearly, there is a marked reduction in specific ORC subunits that is sufficient to impact replication during development and in fibroblasts, but the devil's advocate can always claim minimal levels of ORC remaining in these specialized cells.

The significance of the work is that the authors keep improving their conditional alleles (and combining them), thus making it harder and harder (but not impossible) to invoke limiting but sufficient levels of ORC. This work lays the foundation for future functional screens to identify other factors that may modulate the response to the loss of ORC subunits.

This work will be of interest to the DNA replication, polyploidy, and genome stability communities.

---

## [Referee Report · Reviewer #2 (Public review)]

This manuscript proposes that primary hepatocytes can replicate their DNA without the six-subunit ORC. This follows previous studies that examined mice that did not express ORC1 in the liver. In this study, the authors suppressed expression of ORC2 or ORC1 plus ORC2 in the liver.

Comments:

(1) I find the conclusion of the authors somewhat hard to accept. Biochemically, ORC without the ORC1 or ORC2 subunits cannot load the MCM helicase on DNA. The question arises whether the deletion in the ORC1 and ORC2 genes by Cre is not very tight, allowing some cells to replicate their DNA and allow the liver to develop, or whether the replication of DNA proceeds via non-canonical mechanisms, such as break-induced replication. The increase in the number of polyploid cells in the mice expressing Cre supports the first mechanism, because it is consistent with few cells retaining the capacity to replicate their DNA, at least for some time during development.

(2) Fig 1H shows that 5 days post infection, there is no visible expression of ORC2 in MEFs with the ORC2 flox allele. However, at 15 days post infection, some ORC2 is visible. The authors suggest that a small number of cells that retained expression of ORC2 were selected over the cells not expressing ORC2. Could a similar scenario also happen in vivo?

(3) Figs 2E-G show decreased body weight, decreased liver weight and decreased liver to body weight in mice with recombination of the ORC2 flox allele. This means that DNA replication is compromised in the ALB-ORC2f/f mice.

(4) Figs 2I-K do not report the number of hepatocytes, but the percent of hepatocytes with different nuclear sizes. I suspect that the number of hepatocytes is lower in the ALB-ORC2f/f mice than in the ORC2f/f mice. Can the authors report the actual numbers?

(5) Figs 3B-G do not report the number of nuclei, but percentages, which are plotted separately for the ORC2-f/f and ALB-ORC2-f/f mice. Can the authors report the actual numbers?

(6) Fig 5 shows the response of ORC2f/f and ALB-ORC2f/f mice after partial hepatectomy. The percent of EdU+ nuclei in the ORC2-f/f (aka ALB-CRE-/-) mice in Fig 5H seems low. Based on other publications in the field it should be about 20-30%. Why is it so low here? The very low nuclear density in the ALB-ORC2-f/f mice (Fig 5F) and the large nuclei (Fig 5I) could indicate that cells fire too few origins, proceed through S phase very slowly and fail to divide.

(7) Fig 6F shows that ALB-ORC1f/f-ORC2f/f mice have very severe phenotypes in terms of body weight and liver weight (about on third of wild-type!!). Fig 6H and 6I, the actual numbers should be presented, not percentages. The fact that there are EYFP negative cells, implies that CRE was not expressed in all hepatocytes.

(8) Comparing the EdU+ cells in Fig 7G versus 5G shows very different number of EdU+ cells in the control animals. This means that one of these images is not representative. The higher fraction of EdU+ cells in the double-knockout could mean that the hepatocytes in the double-knockout take longer to complete DNA replication than the control hepatocytes. The control hepatocytes may have already completed DNA replication, which can explain why the fraction of EdU+ cells is so low in the controls. The authors may need to study mice at earlier time points after partial hepatectomy, i.e. sacrifice the mice at 30-32 hours, instead of 40-52 hours.

(9) Regarding the calculation of the number of cell divisions during development: the authors assume that all the hepatocytes in the adult liver are derived from hepatoblasts that express Alb. Is it possible to exclude the possibility that pre-hepatoblast cells that do not express Alb give rise to hepatocytes? For example, the cells that give rise to hepatoblasts may proliferate more times than normal giving rise to a higher number of hepatoblasts than in wild-type mice.

(10) My interpretation of the data is that not all hepatocytes have the ORC1 and ORC2 genes deleted (eg EYFP-negative cells) and that these cells allow some proliferation in the livers of these mice.

My comments regarding the previous version still stand, since the authors did not perform experiments to address them.

---

## [Referee Report · Reviewer #3 (Public review)]

Summary:

The authors address the role of ORC in DNA replication and that this protein complex is not essential for DNA replication in hepatocytes. They provide evidence that ORC subunit levels are substantially reduced in cells that have been induced to delete multiple exons of the corresponding ORC gene(s) in hepatocytes. They evaluate replication both in purified isolated hepatocytes and in mice after hepatectomy. In both cases, there is clear evidence that DNA replication does not decrease at a level that corresponds with the decrease in detectable ORC subunit and that endoreduplication is the primary type of replication observed. It remains possible that small amounts of residual ORC are responsible for the replication observed, although the authors provide arguments against this possibility. The mechanisms responsible for the DNA replication observed in the absence of ORC are not examined, including why such replication would primarily be due to endoreduplication.

Strengths:

The authors clearly show that there are dramatic reductions in the amount of the targeted ORC subunits in the cells that have been targeted for deletion. They also provide clear evidence that there is replication in a subset of these cells and that it is likely due to endoreduplication. Although there is no replication in MEFs derived from cells with the deletion, there is clearly DNA replication occurring in hepatocytes (both isolated in culture and in the context of the liver). Interestingly, the cells undergoing replication exhibit enlarged cell sizes and elevated ploidy indicating endoreduplication of the genome. These findings raise the interesting possibility that endoreduplication does not require ORC while normal replication does.

Weaknesses:

There remain two significant weaknesses in this manuscript. The first is that although there is clearly robust reduction of the targeted ORC subunit, the authors cannot confirm that it is deleted in all cells. For example, the analysis in Fig. 4B would suggest that a substantial number of cells have not lost the targeted region of ORC2. In their response, the authors suggest that this is due to contaminating non-hepatocyte cells but do not provide evidence that this is the case. Although the western blots show stronger effects, this type of analysis is notorious for non-linear response curves and no standards are not provided. The second weakness is that there is no evaluation of the molecular nature of the replication observed. In response to the initial review the authors point out that a previous publication mapped Mcm2-7 loading in the absence of ORC1, ORC2 and ORC5 and saw no deficit or altered location. Unfortunately, this is not done for the mutants discussed here and this previous data supports a model that limiting residual ORC is responsible for the replication observed rather than some novel mechanism (which would be expected to alter location or amounts of loading). The manuscript provides no exploration of why "ORC-independent" replication would drive endoreduplicaiton (which is the strongest evidence for an alternative mechanism of initiation but is unique to this experiment and not the previously mutants analyzed for Mcm2-7 loading). Most importantly, it remains true that after numerous papers from this lab and others claiming that ORC is not required for eukaryotic DNA replication, we still have no information about an alternative pathway that could explain Mcm2-7 loading in the absence of ORC. Without some insights in this area, studies such as these will remain controversial.

---

## [Author Response]

The following is the authors’ response to the original reviews.

**eLife Assessment**
This descriptive manuscript builds on prior research showing that the elimination of Origin Recognition Complex (ORC) subunits does not halt DNA replication. The authors use various methods to genetically remove one or two ORC subunits from specific tissues and observe continued replication, though it may be incomplete. The replication appears to be primarily endoreduplication, indicating that ORC-independent replication may promote genome reduplication without mitosis. Despite similar findings in previous studies, the paper provides convincing genetic evidence in mice that liver cells can replicate and undergo endoreduplication even with severely depleted ORC levels. While the mechanism behind this ORC-independent replication remains unclear, the study lays the groundwork for future research to explore how cells compensate for the absence of ORC and to develop functional approaches to investigate this process. The reviewers agree that this valuable paper would be strengthened significantly if the authors could delve a bit deeper into the nature of replication initiation, potentially using an origin mapping experiment. Such an exciting contribution would help explain the nature of the proposed new type of Mcm loading, thereby increasing the impact of this study for the field at large.

We appreciate the reviewers’ suggestion. Till now we know of only one paper that mapped origins of replication in regenerating mouse liver, and that was published two months back in Cell (PMID: 39293447). We want to adopt this method, but we do not need it to answer the question asked. We have mapped origins of replication in ORC-deleted cancer cell lines and compared to wild-type cells in Shibata et al., BioRXiv (PMID: 39554186) (it is under review). We report the following: Mapping of origins in cancer cell lines that are wild type or engineered to delete three of the subunits, *ORC1*, *ORC2* or *ORC5* shows that specific origins are still used and are mostly at the same sites in the genome as in wild type cells. Of the 30,197 origins in wild type cells (with ORC), only 2,466 (8%) are not used in any of the three ORC deleted cells and 18,319 (60%) are common between the four cell types. Despite the lack of ORC, excess MCM2-7 is still loaded at comparable rates in G1 phase to license reserve origins and is also repeatedly loaded in the same S phase to permit re-replication.

Citation: Specific origin selection and excess functional MCM2-7 loading in ORC-deficient cells. Yoshiyuki Shibata, Mihaela Peycheva, Etsuko Shibata, Daniel Malzl, Rushad Pavri, Anindya Dutta. bioRxiv 2024.10.30.621095; doi: https://doi.org/10.1101/2024.10.30.621095 (PMID: 39554186)

We have now included this in the discussion.

**Public Reviews:**

**Reviewer #1 (Public review):**
The origin recognition complex (ORC) is an essential loading factor for the replicative Mcm2-7 helicase complex. Despite ORC's critical role in DNA replication, there have been instances where the loss of specific ORC subunits has still seemingly supported DNA replication in cancer cells, endocycling hepatocytes, and *Drosophila* polyploid cells. Critically, all tested ORC subunits are essential for development and proliferation in normal cells. This presents a challenge, as conditional knockouts need to be generated, and a skeptic can always claim that there were limiting but sufficient ORC levels for helicase loading and replication in polyploid or transformed cells. That being said, the authors have consistently pushed the system to demonstrate replication in the absence or extreme depletion of ORC subunits.Here, the authors generate conditional ORC2 mutants to counter a potential argument with prior conditional ORC1 mutants that Cdc6 may substitute for ORC1 function based on homology. They also generate a double ORC1 and ORC2 mutant, which is still capable of DNA replication in polyploid hepatocytes. While this manuscript provides significantly more support for the ability of select cells to replicate in the absence or near absence of select ORC subunits, it does not shed light on a potential mechanism.The strengths of this manuscript are the mouse genetics and the generation of conditional alleles of ORC2 and the rigorous assessment of phenotypes resulting from limiting amounts of specific ORC subunits. It also builds on prior work with ORC1 to rule out Cdc6 complementing the loss of ORC1.The weakness is that it is a very hard task to resolve the fundamental question of how much ORC is enough for replication in cancer cells or hepatocytes. Clearly, there is a marked reduction in specific ORC subunits that is sufficient to impact replication during development and in fibroblasts, but the devil's advocate can always claim minimal levels of ORC remaining in these specialized cells.The significance of the work is that the authors keep improving their conditional alleles (and combining them), thus making it harder and harder (but not impossible) to invoke limiting but sufficient levels of ORC. This work lays the foundation for future functional screens to identify other factors that may modulate the response to the loss of ORC subunits.This work will be of interest to the DNA replication, polyploidy, and genome stability communities.

Thank you.

**Reviewer #2 (Public review):**
This manuscript proposes that primary hepatocytes can replicate their DNA without the six-subunit ORC. This follows previous studies that examined mice that did not express ORC1 in the liver. In this study, the authors suppressed expression of ORC2 or ORC1 plus ORC2 in the liver.Comments:(1) I find the conclusion of the authors somewhat hard to accept. Biochemically, ORC without the ORC1 or ORC2 subunits cannot load the MCM helicase on DNA. The question arises whether the deletion in the ORC1 and ORC2 genes by Cre is not very tight, allowing some cells to replicate their DNA and allow the liver to develop, or whether the replication of DNA proceeds via non-canonical mechanisms, such as break-induced replication. The increase in the number of polyploid cells in the mice expressing Cre supports the first mechanism, because it is consistent with few cells retaining the capacity to replicate their DNA, at least for some time during development.

In our study, we used EYFP as a marker for Cre recombinase activity. ~98% of the hepatocytes in tissue sections and cells in culture express EYFP, suggesting that the majority of hepatocytes successfully expressed the Cre protein to delete the *ORC1* or *ORC2* genes. To assess deletion efficiency, we employed sensitive genotyping and Western blotting techniques to confirm the deletion of *ORC1* and *ORC2* in hepatocytes isolated from Alb-Cre mice. Results in Fig. 2C and Fig. 6D demonstrate the near-complete absence of ORC2 and ORC1 proteins, respectively, in these hepatocytes.

The mutant hepatocytes underwent at least 15–18 divisions during development. The inherited ORC1 or ORC2 protein present during the initial cell divisions, would be diluted to less than 1.5% of wild-type levels within six divisions, making it highly unlikely to support DNA replication, and yet we observe hepatocyte numbers that suggest there was robust cell division even after that point.

Furthermore, the EdU incorporation data confirm DNA synthesis in the absence of ORC1 and ORC2. Specifically, immunofluorescence showed that both in vitro and in vivo, EYFP-positive hepatocytes (indicating successful *ORC1* and *ORC2* deletion) incorporated EdU, demonstrating that DNA synthesis can occur without ORC1 and ORC2.

Finally, the Alb-ORC2f/f mice have 25-37.5% of the number of hepatocyte nuclei compared to WT mice (Table 2). If that many cells had an undeleted *ORC2* gene, that would have shown up in the genotyping PCR and in the Western blots.

(2) Fig 1H shows that 5 days post infection, there is no visible expression of ORC2 in MEFs with the ORC2 flox allele. However, at 15 days post infection, some ORC2 is visible. The authors suggest that a small number of cells that retained expression of ORC2 were selected over the cells not expressing ORC2. Could a similar scenario also happen in vivo?

This would not explain the significant incorporation of EdU in hepatocytes that are EYFP positive and do not have detectable ORC by Western blots. Also note that for MEFs we are delivering the Cre by Adenovirus infection in vitro, so there is a finite probability that a cell will not receive the virus, the Cre and will not delete ORC2. However, in vivo, the Alb-Cre will be expressed in every cell that turns on albumin. There is no escaping the expression of Cre.

(3) Figs 2E-G shows decreased body weight, decreased liver weight and decreased liver to body weight in mice with recombination of the ORC2 flox allele. This means that DNA replication is compromised in the ALB-ORC2f/f mice.

It is possible that DNA replication is partially compromised or may slow down in the absence of ORC2. However, it is important to emphasize that livers with ORC2 deletion remain capable of DNA replication, so much so that liver function and life span are near normal. Therefore, some kind of DNA replication has to serve as a compensatory mechanism in the absence of ORC2 to maintain liver function and support regeneration.

(4) Figs 2I-K do not report the number of hepatocytes, but the percent of hepatocytes with different nuclear sizes. I suspect that the number of hepatocytes is lower in the ALB-ORC2f/f mice than in the ORC2f/f mice. Can the authors report the actual numbers?

We show in Table 2 that the *Alb-Orc2f/f* mice have about 25-37.5% of the hepatocytes compared to the WT mice.

(5) Figs 3B-G do not report the number of nuclei, but percentages, which are plotted separately for the ORC2-f/f and ALB-ORC2-f/f mice. Can the authors report the actual numbers?

In all the FACS experiments in Fig. 3B-G we collect data for a total of 10,000 nuclei (or cells). For Fig. 3E-G we divide the 10,000 nuclei into the bottom 40% on the EYFP axis (EYFP low, which is mostly EYFP negative) as the control group, and EYFP high (top 20% on the EYFP axis) test group. We have described this in the Methods in the revision and labeled EYFP negative and positive as EYFP low and high in the Figures and Figure legends.

(6) Fig 5 shows the response of ORC2f/f and ALB-ORC2f/f mice after partial hepatectomy. The percent of EdU+ nuclei in the ORC2-f/f (aka ALB-CRE-/-) mice in Fig 5H seems low. Based on other publications in the field it should be about 20-30%. Why is it so low here? The very low nuclear density in the ALB-ORC2-f/f mice (Fig 5F) and the large nuclei (Fig 5I) could indicate that cells fire too few origins, proceed through S phase very slowly and fail to divide.

The percentage of EdU+ nuclei in the *ORC2f/f* without Alb-Cre mice is 8%, while in PMID 10623657 ~10% of wild type nuclei incorporate EdU at 42 hr post partial hepatectomy (mid-point between the 36-48 hr post hepatectomy that was used in our study). The important result here is that in the *ORC2f/f* mice with Alb-Cre (+/-) we are seeing significant EdU incorporation. We have also corrected the X-axis labels in 5F, 5I, 7E and 7F to reflect that those measurements were not made at 36 hr post-resection but later (as was indicated in the schematic in Fig. 5A).

(7) Fig 6F shows that ALB-ORC1f/f-ORC2f/f mice have very severe phenotypes in terms of body weight and liver weight (about on third of wild-type!!). Fig 6H and 6I, the actual numbers should be presented, not percentages. The fact that there are EYFP negative cells, implies that CRE was not expressed in all hepatocytes.

The liver weight is very dependent on the body weight, and so we have to look at the liver to body weight ratio to determine if it is inordinately small, and the ratio is 70% of the WT. In females the liver and body weight are low (although in proportion to each other), which maybe is what the reviewer is talking about. However, the fact that liver weight and body weight are not as low in males, suggest that this is a gender (hormone?) specific effect and not a DNA replication defect. We had discussed this possibility. We have another paper also in BioRXiv (Su et al. doi.org/10.1101/2024.12.18.629220) that suggests that ORC subunits have significant effect on gene expression, so it is possible that that is what leads to this sexual dimorphism in phenotype. We have now added this to the discussion.

The bottom 40% of nuclei on the EYFP axis in the FACS profiles (what was labeled EYFP negative but will now be called EYFP low) contains mostly non-hepatocytes that are genuinely EYFP negative. Non-hepatocytes (bile duct cells, endothelial cells, Kupffer cells, blood cells) are a significant part of cells in the dissociated liver (as can be seen in the single cell sequencing results in PMID: 32690901). Their presence does not mean that hepatocytes are not expressing Cre. Hepatocytes are nearly 100% EYFP positive, as can be seen in the tissue sections (where the hepatocytes take up most of visual field) and in cells in culture. Also if there are EYFP negative hepatocyte nuclei in the FACS, that still does not rule out EYFP presence in the cytoplasm. The important point from the FACS is that the EYFP high nuclei (which have expressed Cre for the longest period) are polyploid relative to the EYFP low nuclei.

(8) Comparing the EdU+ cells in Fig 7G versus 5G shows very different number of EdU+ cells in the control animals. This means that one of these images is not representative. The higher fraction of EdU+ cells in the double-knockout could mean that the hepatocytes in the double-knockout take longer to complete DNA replication than the control hepatocytes. The control hepatocytes may have already completed DNA replication, which can explain why the fraction of EdU+ cells is so low in the controls. The authors may need to study mice at earlier time points after partial hepatectomy, i.e. sacrifice the mice at 30-32 hours, instead of 40-52 hours.

The apparent difference that the reviewer comments on stems from differences in nuclear density in the images in Fig. 7G and 5G (also quantitated in Fig. 7F and 5F). The quantitation in Fig. 7H and 5H show that the % of EdU plus cells are comparable (5-8%).

(9) Regarding the calculation of the number of cell divisions during development: the authors assume that all the hepatocytes in the adult liver are derived from hepatoblasts that express Alb. Is it possible to exclude the possibility that pre-hepatoblast cells that do not express Alb give rise to hepatocytes? For example the cells that give rise to hepatoblasts may proliferate more times than normal giving rise to a higher number of hepatoblasts than in wild-type mice.

Single cell sequencing of mouse liver at e11 shows hepatoblasts expressing hepatocyte specific markers (PMID: 32690901). All the cells annotated from the single-cell seq analysis are differentiated cells arguing against the possibility that undifferentiated endodermal cells (what the reviewer probably means by pre-hepatoblasts) exist at e11. We have added this citation to the paper.

Here is a review that says the hepatoblasts expressing Albumin are present before e13. (https://www.ncbi.nlm.nih.gov/books/NBK27068/) says: “The differentiation of bi-potential hepatoblasts into hepatocytes or BECs begins around e13 of mouse development. Initially hepatoblasts express genes associated with both adult hepatocytes (*Hnf4α, Albumin*) ...” Thus, we can be certain that hepatoblasts before e13 express albumin. Our calculation of number of cell divisions in Table 2 begins from e12.

The reviewer may be suggesting that ORC deletion leads to the immediate demise of hepatoblasts (despite having inherited ORC protein from the endodermal cells) causing undifferentiated endodermal cells to persist and proliferate much longer than in normal development. We consider it unlikely, but if true it will be very unexpected, both by suggesting that deletion of ORC immediately leads to the death of the hepatoblasts (despite a healthy reserve of inherited ORC protein) and by suggesting that there is a novel feedback mechanism from the death/depletion of hepatoblasts leading to the persistence and proliferation of undifferentiated endodermal cells. We have added the reviewer’s suggestion to the discussion.

(10) My interpretation of the data is that not all hepatocytes have the ORC1 and ORC2 genes deleted (eg EYFP-negative cells) and that these cells allow some proliferation in the livers of these mice.

Please see the reply in question #1. Particularly relevant: “Finally, the *Alb-ORC2f/f* mice have 25-37.5% of the number of hepatocyte nuclei compared to WT mice (Table 2). If that many cells had an undeleted ORC2 gene, that would have shown up in the genotyping PCR and in the Western blots.

**Reviewer #3 (Public review):**
Summary:The authors address the role of ORC in DNA replication and that this protein complex is not essential for DNA replication in hepatocytes. They provide evidence that ORC subunit levels are substantially reduced in cells that have been induced to delete multiple exons of the corresponding ORC gene(s) in hepatocytes. They evaluate replication both in purified isolated hepatocytes and in mice after hepatectomy. In both cases, there is clear evidence that DNA replication does not decrease at a level that corresponds with the decrease in detectable ORC subunit and that endoreduplication is the primary type of replication observed. It remains possible that small amounts of residual ORC are responsible for the replication observed, although the authors provide arguments against this possibility. The mechanisms responsible for DNA replication in the absence of ORC are not examined.Strengths:The authors clearly show that there are dramatic reductions in the amount of the targeted ORC subunits in the cells that have been targeted for deletion. They also provide clear evidence that there is replication in a subset of these cells and that it is likely due to endoreduplication. Although there is no replication in MEFs derived from cells with the deletion, there is clearly DNA replication occurring in hepatocytes (both isolated in culture and in the context of the liver). Interestingly, the cells undergoing replication exhibit enlarged cell sizes and elevated ploidy indicating endoreduplication of the genome. These findings raise the interesting possibility that endoreduplication does not require ORC while normal replication does.Weaknesses:There are two significant weaknesses in this manuscript. The first is that although there is clearly robust reduction of the targeted ORC subunit, the authors cannot confirm that it is deleted in all cells. For example, the analysis in Fig. 4B would suggest that a substantial number of cells have not lost the targeted region of ORC2. Although the western blots show stronger effects, this type of analysis is notorious for non-linear response curves and no standards are provided. The second weakness is that there is no evaluation of the molecular nature of the replication observed. Are there changes in the amount of location of Mcm2-7 loading that is usually mediated by ORC? Does an associated change in Mcm2-7 loading lead to the endoreduplication observed? After numerous papers from this lab and others claiming that ORC is not required for eukaryotic DNA replication in a subset of cells, we still have no information about an alternative pathway that could explain this observation.

We do not see a significant deficit in MCM2-7 loading (amount and rate) in cancer cell lines where we have deleted *ORC1*, *ORC2* or *ORC5* genes separately in Shibata et al. bioRxiv 2024.10.30.621095; doi: https://doi.org/10.1101/2024.10.30.621095 (PMID: 39554186). This is now cited in the discussion.

The authors frequently use the presence of a Cre-dependent eYFP expression as evidence that the ORC1 or ORC2 genes have been deleted. Although likely the best visual marker for this, it is not demonstrated that the presence of eYFP ensures that ORC2 has been targeted by Cre. For example, based on the data in Fig. 4B, there seems to be a substantial percentage of ORC2 genes that have not been targeted while the authors report that 100% of the cells express eYFP.

(1) The PCR reactions in Fig. 4B are still contaminated by DNA from non-hepatocyte cells: bile duct cells, endothelial, Kupfer cells and blood cells. Microscopy of cultured cells idnetifies the hepatocytes unequivocally from their morphology. <2% of the hepatocyte cells in culture in Fig. 4C are EYFP-.

**Recommendations for the authors:**

**Reviewer #2 (Recommendations for the authors):**
The authors should present the data as suggested in the review and reformulate their conclusions. If possible, mice should be examined 30-32 hours after partial hepatectomy.

Based on the Literature we chose a time that is consistent with the previous paper from us (Uchida et al., Genes & Dev).

**Reviewer #3 (Recommendations for the authors):**
(1) It would improve the paper to use single-cell methods (e.g. FISH) to assess the deletion of ORC subunits in the targeted cells.

This is something we will reserve for future studies.

(2) The importance of the paper would be increased dramatically by showing that the elimination of ORC changed the location of Mcm2-7 loading. This would be highly likely if the authors hypothesis that ORC is not involved is true. On the other hand, given ORC's role in origin selection, an observation that the same sites are used but less frequently would support a hypothesis that residual intact ORC is responsible for the replication observed.

Shibata et al (PMID: 39554186) has answered this question. The loss of ORC does not change the locations of origins or even the ability to specify origins. We argue that this is what is to be expected from our hypothesis, that although ORC is clearly important for MCM loading in yeast and in biochemical experiments, something unexpected is going on in human cells. Either a vanishingly small amount of ORC (undetectable by commonly used methods) can load the full complement of MCM2-7 at a rate that is comparable to wild type cells, or there is an ORC-independent mechanism of MCM2-7 loading. This is now added to the discussion.